# IntelliAsk: Learning to Ask Critical Questions with Human-Aligned Rewards

## Abstract

Peer review relies on substantive, evidence-based questions, but existing LLM-based approaches often generate surface-level queries. We find that LLM-generated questions take over 50% of their question tokens from a paper's first page, while human reviewers draw on the full text. Human questions are also more insightful, showing effort and grounding, whereas LLM questions mostly reflect surface style. To address this, we extract 151k candidate questions from ICLR 2024 reviews and filter them through a multi-stage filtering process into Probe-15K, a set of 15.5k high-quality questions. From this, we create ProbeVote-500, where human annotators score questions along effort, evidence, and grounding. Using these labels, we train IntelliReward, a reward model built from a frozen Autoregressive LLM with trainable multi-head transformers over the final 50 token states. This architecture outperforms API-based SFT finetuning (Gemini 2.5 Flash, GPT-4.1) as baselines for reward. Applying DAPO with IntelliReward, we train IntelliAsk, a question-generation model aligned with human preferences and substantially stronger than existing fine-tuned review models. Finally, by releasing Probe-15K, ProbeVote-500, and IntelliReward, we provide an automatic evaluation benchmark for reviewer questions that measures groundedness, effort, and evidence. We release our implementation and datasets to ensure reproducibility (code available at https://anonymous.4open.science/r/IntelliA-3E09/).

## 1 Introduction

Asking critical and well-reasoned questions is essential for advancing research, as such questions help clarify ideas, reveal limitations, and inspire new directions. In academic publishing, peer review plays a key role in this process, relying on reviewers to raise questions that improve the quality and impact of scientific work. However, as the number of submissions to major conferences has grown, the quality of reviewer feedback has declined. Many reviewers are overloaded and face tight deadlines, leading some to rely on large language models (LLMs) to draft questions and comments (Liang et al., 2024). While LLMs can produce fluent text, the questions they generate often lack technical depth, proper reasoning, or contextual understanding of the work.

**Why existing resources are not enough.** Most of the recent research works propose methods to improve the review generation capabilities of the LLMs. However, there's no focus on the quality of critic and the questions in the review generated by the models trained using these techniques, hence rendering the review useless. Closer to our setting, Idahl & Ahmadi (2025), fine-tunes LLaMA-8B on 79k reviews, but the generated questions extracted from the peer review just mimic the tone of reviewer style (See Section 4). The generated questions "sound" human, without offering a comprehensible and thoughtful question. Chitale et al. (2025) uses a Graph based approach for generating peer reviews. While the graph structure helps organize paper content, the model still relies on simple supervised fine-tuning and produces questions that lack critical depth, remaining shallow imitations of human phrasing. Moreover, both Idahl & Ahmadi (2025) & Chitale et al. (2025) evaluate their systems primarily with automated review-quality scores from LLM judges, without incorporating human-in-the-loop assessments to measure whether the questions are actually useful to authors. Similarly, Dasigi et al. (2021) uses only titles and abstracts to generate questions, limiting the scope for creating technically detailed peer questions that are meaningful to authors. Overall, these approaches frame the task too broadly - treating it as generic review or QA generation-without explicitly modeling what makes reviewer questions effortful, evidence-based, and grounded.

**Challenges.** Generating effective review questions is not the same task as producing generic QA pairs based on the available content. LLM-generated questions often lack a clear understanding of technical content, resulting in questions that may be verbose and lengthy but unhelpful or already answered in the paper. Our own experiments highlight this gap: we conducted an experiment where four expert annotators evaluated the questions generated by 3 strong baseline LLMs. They rated four variants of questions (3 model-generated and 1 human-written question from Openreview) each from o3, Gemini 2.5 Pro, Qwen2.5-32B and compared them to real human-authored questions. When evaluated with our rubric, **humans scored 0.78 points higher on average than the strongest model** and **1.53 points higher on average than the lowest scoring model** (see Table 3). The results show that human-written questions were consistently more relevant and useful. They were categorized to be written with more effort, contained evidence from the paper and weren't just framed using keywords from the paper, while the converse was true for the questions asked by the LLMs.

**Our Work.** In this paper, we address the challenge of generating critical, well-reasoned review questions. We curate **Probe-15K**, a dataset of 15,500 reviewer questions from ICLR 2024 submissions, filtered for technical depth, clarity, and diversity. We then introduce **ProbeVote-500**, an expert-annotated set of question–paper pairs scored on Effort, Evidence, and Grounding, and use it to train **IntelliReward**, a reward model that serves as a scalable evaluation benchmark aligned with expert judgments. Finally, we show that while supervised fine-tuning (SFT) mostly imitates reviewer style, reinforcement learning (RL with DAPO) guided by IntelliReward achieves closer alignment with human-authored questions.

Our contributions are as follows:

1. **Probe-15K dataset**: A rigorously filtered collection of 15,500 reviewer questions drawn from 151K OpenReview entries for ICLR 2024, via multi-stage filtering- assessing substance, effort, and grounding, to arrive at 15K questions.

2. **ProbeVote-500 and IntelliReward**: ProbeVote-500, A dataset of expert-annotated question–paper pairs using three criteria - Effort, Evidence, and Grounding. From it, we build IntelliReward, an auto-eval benchmark that aligns more closely with human judgment and outperforms API based LLM-as-judge baselines tuned using SFT. To validate our reward model, we trained a 7B and 32B Model using IntelliReward for quality critical question generation.

## 2 PROBE-15K

### 2.1 LARGE-SCALE EXTRACTION OF QUESTIONS FROM OPENREVIEW REVIEWS

We collected a dataset of reviewer feedback by scraping all publicly available reviews from ICLR 2024 using the OpenReview API. For each paper, we retrieved the corresponding metadata and downloaded the main PDF (excluding supplementary materials), limiting the maximum length to nine pages.

An Openreview submission includes several structured fields: *Summary*, *Strengths*, *Weaknesses*, *Questions*, *Limitations*, *Ethical Concerns*, numerical scores for *Soundness* and *Overall Evaluation*, and the reviewer's *Confidence*. In practice, however, reviewers do not consistently confine their questions to the *Questions* field. To characterize variability in question placement, we manually annotated a random sample of 100 reviews, observing that questions frequently appeared outside the designated *Questions* section, sometimes they are present within the *Weaknesses* or, less frequently, the *Strengths* (See Fig12 in A.9). In some cases, the *Questions* section points to other sections (e.g., "See Weaknesses"), or mixed multiple questions with commentary.

To address this variability and extract reviewer questions, we used Gemini 2.0 and prompted it with the concatenated text of the *Questions*, *Strengths*, and *Weaknesses* sections from each review. The prompt explicitly instructed the model to copy questions verbatim, preserving their original phrasing and tone. (see A.13.4 in A for the full prompt). When a reviewer wrote multiple independent queries in a single sentence, the model split them into separate entries. To verify the accuracy, we manually inspected 500 extracted questions to ensure that the model consistently retained the original phrasing and did not hallucinate content.

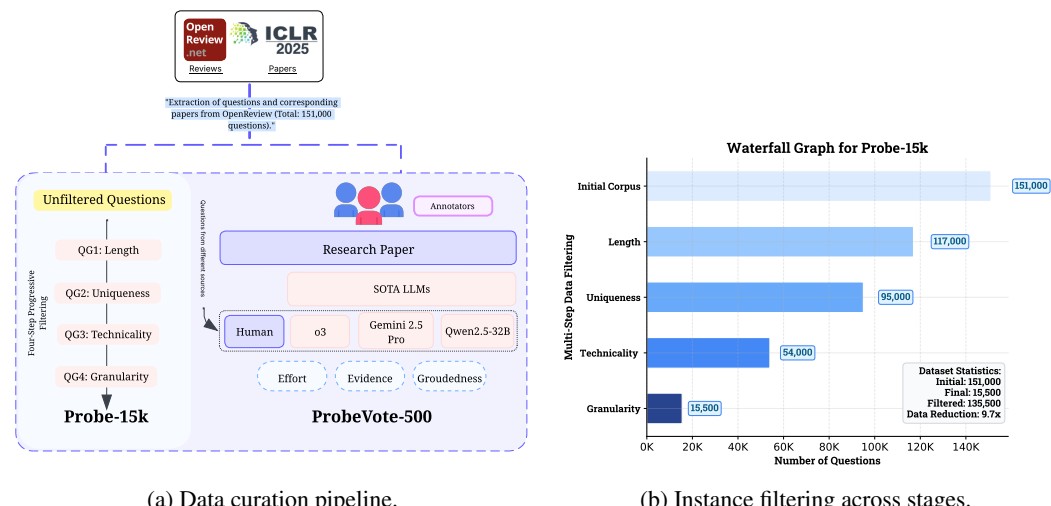

(a) Data curation pipeline.

(b) Instance filtering across stages.

Figure 1: Overview of the dataset construction process: (a) multi-stage data filtering steps for curation, and (b) waterfall diagram illustrating progressive filtering at each stage

## 2.2 CONSTRUCTING PROBE-15K AND ITERATIVE QUALITY ENHANCEMENT

To construct a dataset suitable for downstream modeling, we applied a series of filtering steps guided by best practices from CVPR reviewer slides (Davis, 2021), NeurIPS (NeurIPS, 2023) and ICLR reviewer guidelines (ICLR, 2025), prior work on LLM feedback for reviews (Thakkar et al., 2025), and our own manual inspection of roughly 2,000 reviews. The initial extraction produced about 151,000 questions. Our goal was not simply to maximize quantity but to ensure that the retained questions were clear, specific, and technically relevant. Each filtering stage systematically removed low-quality or redundant entries. After every stage, we manually checked a random sample of about 1,000 questions to confirm that the filtering criteria were effective and that valid questions were not being discarded.

**Length-Based Filtering.** We first excluded questions under 100 characters. Manual analysis showed that short questions typically contained superficial comments or clarifications readily apparent in the submission text. This filtering step removed 34,000 entries, resulting in a subset of 117,000 questions. We then proceed to remove semantically similar questions.

**Eliminating Semantically Redundant Questions.** Numerous questions were semantically identical apart from minor variations in wording. Training on highly redundant content increases the risk of overfitting and limits output diversity. To address this, we applied clustering using Stella with a cluster size of k=5. This reduced the dataset to 95,000 questions. After this stage of filtering there were still many questions which were non-technical and not relevant to the content of the paper for which we employ another stage of filtering described further.

**Filtering Non-Technical and Irrelevant Content.** Manual review identified many questions unrelated to the technical content, including remarks on grammar, formatting, typographic errors, and unprofessional or subjective comments. Prior work (Liang et al., 2024) has shown that reviews containing certain keywords (e.g., "commendable," "innovative") are often generated by language models. To mitigate this, we developed a prompt specifying six exclusion criteria, detailed in the Appendix(See A.13.1). Importantly, we provided Gemini 2.0 Flash with both the review text and the corresponding paper as context, ensuring that ungrounded or off-topic questions could be more reliably detected and filtered. This process removed 41,000 questions. Even after this stage, we observed remaining questions that were purely opinion-based or that dismissed techniques without justification, which were addressed in the subsequent filtering stage.

**Filtering for Specificity and Actionability.** The final stage removed questions that were vague or speculative. We targeted two categories: (i) incomplete, rhetorical, or opinion-based questions without supporting evidence; (ii) unsupported assertions that a technique would fail or had been

previously published (See A.13.2 in A). Questions were sequentially evaluated, retaining only those that satisfied all criteria. This step removed 38,500 questions, resulting in a final corpus of approximately 15,500 diverse, technically relevant entries.

After filtering, the final dataset contained 15.5k questions drawn from 5,841 unique papers. The train dataset contains 13.2k questions and the test dataset contains 2.3k questions. To prepare the corresponding paper content for evaluation and training, we applied `olmOCR(allenai/olmOCR-7B-0825-FP8)`(Poznanski et al., 2025)to extract structured text from the first nine pages of each paper.

## 3    BENCHMARKING SOTA REASONING LLMS AGAINST HUMANS

LLMs are capable of generating reviews when provided with a complete paper, however, where they tend to fall short is in asking compelling questions that involve critical thinking about the content of the paper and as well as the domain knowledge of the paper under consideration. To study this, we conduct a human annotation study comparing questions extracted from OpenReview reviews with those generated by several state-of-the-art LLMs.

We primarily do this for below two reasons:

1. To benchmark and quantify the gap between human and LLM-generated questions
2. To create the preference data required to train a reward model to scale annotation.

### 3.1    PROBEVOTE-500: HUMAN PREFERENCE AND ANNOTATION DATA

**Experimental Setup.** ProbeVote-500 consists of 572 annotated question–paper pairs abstracted from 300 randomly sampled ICLR 2025 submissions on Openreview. For each paper, the full text was provided as input to the following large language models : Gemini 2.5 Flash (Reasoning model), o3 (Reasoning model), Qwen2.5-32B , under an identical prompting template (see A.13.3), yielding one model-generated question per system. In parallel, the corresponding human-authored reviewer question from Openreview was included as the reference. To eliminate source bias, all questions were anonymized before annotation. Human evaluators read each paper in full, including text, figures, and equations, to ensure proper context (See A.10 for the User-Interface used by Annotators). If a paper was entirely outside an annotator's domain expertise, it was marked as skipped and reassigned. Annotators then scored each anonymized question according to the rubric introduced in Section 3.2, which evaluates three binary dimensions: Effort, Evidence, and Grounding.

### 3.2    RUBRICS FOR ASSESSING QUESTION QUALITY: EFFORT, EVIDENCE, AND GROUNDING

To evaluate question quality, we design a rubric with three binary metrics: Effort, Evidence, and Grounding. Each metric is scored as 0/1, keeping the evaluation simple and consistent across annotators. We chose a binary scheme to reduce ambiguity and to focus on whether a question meets the essential qualities of being thoughtful and useful for authors. See A.11 for examples of each category.

1. **Effort**: Does the question demand real thought to answer? Low-effort questions can be answered by directly quoting the paper or restating surface-level details, whereas a high-effort question requires the reader to synthesize ideas, connect sections, or identify non-obvious implications beyond what is stated.
2. **Evidence**: Is the question backed by specific content from the paper? High-evidence questions point to particular results, assumptions, or arguments in the work and probe them critically. Low-evidence questions raise points without support, making them speculative or unhelpful.
3. **Grounding**: Is the question anchored in the actual content of the paper? Grounded questions refer to concrete methods, experiments, or claims across sections of the paper. Ungrounded questions rely only on generic phrasing, keywords, or broad statements that could apply to almost any paper. For example: What if we increase the depth of the neural network ?

## 3.3 ANALYSIS FROM PROBEVOTE-500

**Source Vs Score.** Blind annotation results show that the Qwen2.5-32B model received the lowest scores, while highest quality human-authored questions from Openreview achieved the highest (see Table 3). The mean cumulative score is calculated by taking an average of all the axis of the rubric, with the highest possible score being 3 and lowest 0. This gap becomes even clear when looking at the specific categories scores in Fig 3.

**Keyword Coverage.** We measure keyword coverage as the fraction of words in the question that originate from the paper's first page. This tests whether models rely disproportionately on introductory text when framing questions. A high score indicates surface-level dependence, while lower scores suggest engagement with the full paper. Qwen2.5-32B shows the strongest dependence, with **55%** of question words coming from the first page alone (Table 3). In contrast, Human-authored questions, o3, and Gemini 2.5 Pro achieve relatively low scores, indicating that they draw more evenly from later sections of the paper when constructing questions.

**Question Length vs Source.** Figure 2 compares question lengths across sources. Qwen2.5-32B produces the shortest questions, while Gemini 2.5 Pro generates the longest. The average length of o3's questions is close to that of Human-authored ones, but Humans show the highest variance, reflecting greater diversity and less reliance on fixed phrasing patterns.

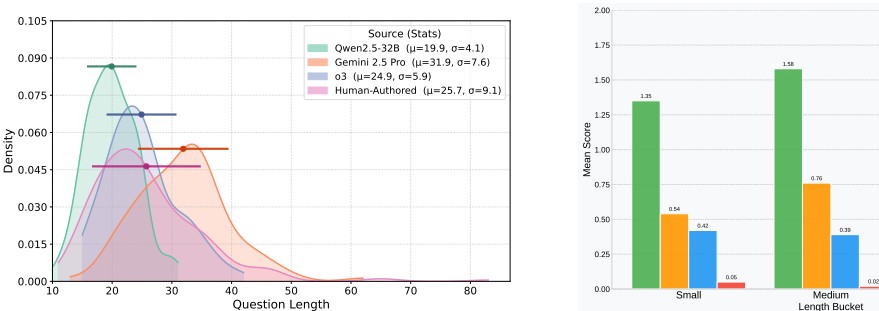

Figure 2: (Left) Distribution of question lengths across sources. Kernel density estimates are shown for human-authored questions and those generated by Qwen2.5-32B, o3, and Gemini 2.5 Pro. (Right) Mean Score Distribution (0-3) vs Length Bucket for different sources

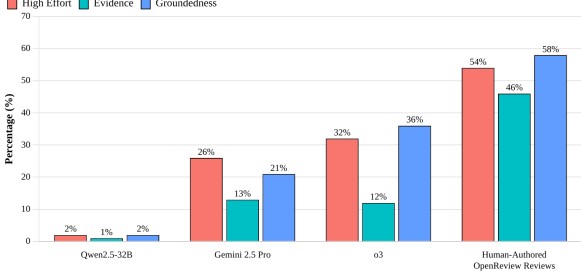

Figure 3: The figures show the distribution of votes on Effort, Evidence and Factual metrics for various sources of questions.

**Question Length vs Score.** Comparing Human-authored questions with o3 reveals clear gaps in quality. For short questions ($< 20$ characters), Human-authored ones are more than **2× richer** in quality (effort + evidence + groundedness) than those from o3. The largest gap is in groundedness, where Humans outperform o3 by over **10×**. Effort is also substantially lower for o3, suggesting that even its concise questions often lack depth and framing.

## 4  WHY SFT FAILS ON THIS TASK

We fine-tuned *Qwen/Qwen2.5-7B-Instruct-1M* on Probe-15K using Human-authored questions as the reference for reviewer-style generation. Training ran on four H200 GPUs for 24 hours with an input length of 14K tokens per paper. For evaluation, we held out a test split of 2200 samples from Probe-15K and used the same prompts as in our human annotation study to ensure fairness.

The fine-tuned model learned to mimic the phrasing and tone of reviewers but did not improve in producing meaningful questions: depth, reasoning, and grounding remained weak compared to Human-authored questions (see Table 3). We also tested existing SFT-trained reviewer models (OpenReviewer, DeepReviewer, AutoRev) by extracting the *Questions* section of their outputs. Their results were fluent in style but shallow in substance, lacking the critical depth of Human-written questions (See A.1).

These findings show that SFT captures style but not reasoning. High-quality reviewer questions require more than surface imitation, motivating our next step: reinforcement learning with IntelliReward, a reward model trained to capture human preferences along Effort, Evidence, and Grounding.

## 5  TRAINING INTELLIASK: A SPECIALIZED MODEL FOR ASKING CRITICAL QUESTIONS

As shown in Section 3 and table 3, SFT does not improve the model's performance on the critical question generation task. This limitation is consistent with recent findings showing that SFT often memorizes training data and struggles with out-of-distribution scenarios. Because of this tendency, it struggles to adapt to new situations. Reinforcement learning (RL), on the other hand, encourages exploration and learning from feedback, which helps it generalize better and handle tasks that require complex reasoning (Chu et al., 2025).

### 5.1  REWARD MODEL : INTELLIREWARD

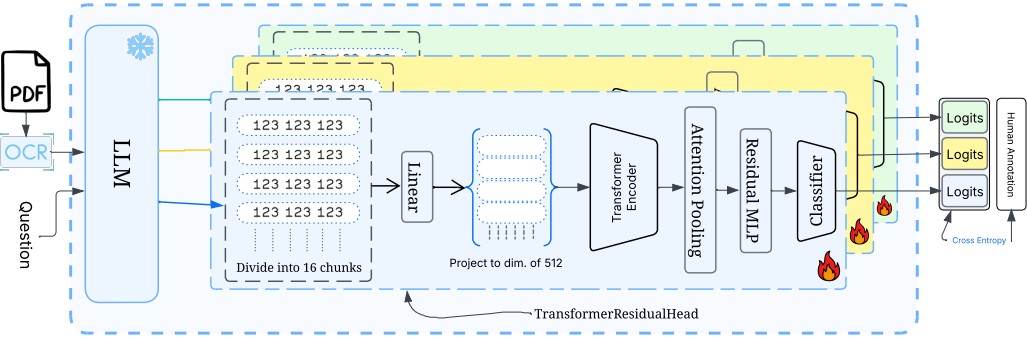

Figure 4: Architecture and training of the IntelliReward .

Evaluating all 15,500 questions with human annotators across three rubrics is costly and risks bias from fatigue. This highlights the need for a reliable automatic evaluation benchmark to support the scaling of our experiments. To reduce reliance on manual effort, we tested leading closed-source LLMs on the reward prediction task. However, they showed weak predictive accuracy (Table 1), required large inputs, and incurred high inference costs, making them unsuitable for large-scale benchmarking. To overcome this, we trained IntelliReward on ProbeVote-500 to serve as an efficient and scalable substitute for human judgment. The architecture and training procedure are described in the following subsection.

### 5.2  REWARD MODEL ARCHITECTURE AND TRAINING

**Reward Model Architecture.**  Our reward model handles multiple objectives by pairing a causal LLM with per-objective Transformer heads. We use gpt-oss-20b (medium reasoning) as the

Table 1: Reward prediction accuracy of different models. We compare off-the-shelf closed-source LLMs, the same models tuned with SFT using their APIs, an open-source baseline (Qwen2.5-7B-Instruct-1M), and our trained model IntelliReward on the test split of ProbeVote-500.

| Candidate reward model | Checkpoint | Effort (%) | Evidence (%) | Grounded (%) | Mean Accuracy(%) |
|---|---|---|---|---|---|
| *Closed-source LLMs (off-the-shelf)* | | | | | |
| Gemini 2.5 Flash | Original/API | 57 | 25 | 29 | 37 |
| GPT-4.1 | Original/API | 44 | 22 | 30 | 32 |
| GPT-5 | Original/API | 56 | 54 | 49 | 53 |
| *Closed-source LLMs (tuned with SFT via API)* | | | | | |
| Gemini 2.5 Flash | SFT/API | 61 | 53 | 45 | 53 |
| GPT-4.1 | SFT/API | 52 | 25 | 31 | 36 |
| *Open-source baseline* | | | | | |
| Qwen2.5-7B-Instruct-1M | Original | 30 | 26 | 28 | 28 |
| *Our trained reward model* | | | | | |
| **IntelliReward (ours)** | – | **70** | **76** | **70** | **72** |

base. Given an input (e.g., paper OCR, generated question, task prompt), the LLM encodes it into a fixed representation. We extract the pooled hidden states of the last 50 output tokens and pass it to our per-objective Transformer head, which empirically improves performance as compared to using MLP head. (see Table 2). The resulting representation is denoted as

$$r \in \mathbb{R}^H, \quad H = 2880,$$

where $r$ is the pooled hidden representation of the LLM outputs and $H$ is its dimensionality.

Each evaluation objective $j \in \{1, \ldots, k\}$ has an independent head $f_j(\cdot)$ producing logits $\ell_j \in \mathbb{R}^{C_j}$, where $k$ is the total number of objectives and $C_j$ is the number of classes (or possible labels) for objective $j$. Each `TransformerResidualHead` first chunks $r$ into $n$ segments and projects them to dimension $d_{\text{model}}$, then processes the sequence through $L$ Transformer encoder layers. A learnable attention query pools the sequence into a vector $z \in \mathbb{R}^{d_{\text{model}}}$, which is refined via a residual two-layer feedforward network (MLP):

$$z' = \text{LayerNorm}\big(z + \text{FFN}(z)\big),$$

where $\text{FFN}(\cdot)$ is the feedforward transformation and $\text{LayerNorm}(\cdot)$ denotes layer normalization. Finally, the refined vector is mapped to logits:

$$\ell_j = W_j z' + b_j,$$

where $W_j \in \mathbb{R}^{C_j \times d_{\text{model}}}$ and $b_j \in \mathbb{R}^{C_j}$ are learnable weights and biases for head $j$.

**Training Objective and Inference** During training, the model minimizes the total loss $\mathcal{L} = \sum_{j=1}^{k} \text{CE}(\ell_j, y_j)$, where CE denotes cross-entropy and $y_j$ is the ground-truth label for objective $j$. During inference, each head predicts $\hat{y}_j = \arg\max \ell_j$, and the final score is computed as $S = \sum_{j=1}^{k} \hat{y}_j$.

**Reward Model Training.** We train IntelliReward using preference annotations from ProbeVote-500. The frozen LLM provides representations, while only the per-objective heads $f_j(\cdot)$ are updated. Training follows the cross-entropy loss defined above. We optimize with AdamW (learning rate $2 \times 10^{-5}$, batch size 8, weight decay 0.01) for 5 epochs on a single NVIDIA L40S GPU. End-to-end training completes within 30 minutes. The Per-objective Head is lightweight and only takes total of 300MB of GPU VRAM during inference.

### 5.3 RL USING INTELLIREWARD REWARD MODEL

As shown in Section 4, supervised fine-tuning (SFT) performs poorly for review question generation: the model copies surface style but does not produce questions with real effort, evidence,

Table 2: Ablation study comparing **MLP** vs. **TransformerResidualHead** (Ours). We group results by head architecture to show the impact of using MLP vs TransformerResidualHead, Pool-k = mean pooling over last k output tokens.

| Configuration | | Accuracy Metrics (%) | | | |
|---|---|---|---|---|---|
| Base Model | Pooling | Effort | Evidence | Grounding | Mean |
| *Head Architecture: Standard MLP* | | | | | |
| Frozen | None | 61 | 64 | 61 | 62 |
| Frozen | Pool50 | 64 | 67 | 64 | 65 |
| Trainable | None | 64 | 65 | 60 | 63 |
| Trainable | Pool50 | 65 | 69 | 67 | 67 |
| Trainable | Pool128 | 64 | 68 | 66 | 66 |
| *Head Architecture: Transformer Residual (Ours)* | | | | | |
| Frozen | None | 68 | 68 | 70 | 69 |
| Frozen | Pool50 | 70 | 76 | 70 | 72 |
| Frozen | Pool128 | 69 | 77 | 67 | 71 |
| Trainable | None | 71 | 69 | 70 | 70 |
| Trainable | Pool50 | 71 | 78 | 70 | **73** |
| Trainable | Pool128 | 70 | 78 | 68 | 72 |

or grounding. To address this, we use our reward model, **IntelliReward**, to align generation with human preferences. Fig 5 shows the difference in reward curve for both Qwen2.5-7B-1M and IntelliAsk.

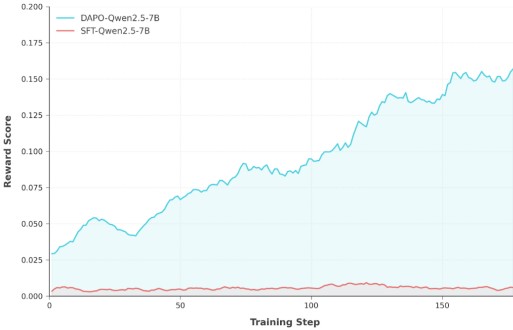

Figure 5: The figures show the difference in reward curves for Qwen2.5-7B (SFT) and IntelliAsk during training.

We train with the DAPO(Yu et al., 2025) for IntelliAsk-7B and using GRPO for IntelliAsk-32B algorithm: for each paper, the model generates several candidate questions, which are scored by IntelliReward, and these scores are used as rewards to guide optimization. Training follows the standard DAPO and GRPO setup (batch sizes, sequence length, gradient clipping, learning rate schedule; see Appendix A.12). The resulting model, **IntelliAsk-32B**, consistently outperforms SFT-only baselines (See table 3) by producing questions that are more evidence-based, better grounded, and require greater effort. IntelliAsk-32B also shows strong gains on external benchmarks.

## 6 RELATED WORK

Recent research has increasingly explored the use of large language models (LLMs) to automate aspects of peer review. Several works train models on large corpora of reviews, often through super-

Table 3: Comparison of reward-model evaluated scores and human-evaluated scores. **Bold** indicates best performance; underline indicates second best. Component scores (Effort, Evidence, Grounding) range from 0 to 1. Total score ranges from 0 to 3 (sum of components). For Coverage, **lower** percentages (↓) indicate better performance.

| Model / Source | Reasoning | Scores [0–1] | | | Total [0–3] | Coverage |
| | | Effort | Evidence | Grounding | | (%) ↓ |
|---|---|---|---|---|---|---|
| *Reward-Model Evaluated: Large Models* | | | | | | |
| gpt-oss-120b | Medium | 0.08 | 0.15 | 0.12 | 0.35 | 22.99 |
| gpt-4.1 | No | 0.07 | 0.12 | 0.12 | 0.31 | 31.73 |
| gpt-5 | Default | 0.09 | **0.20** | 0.16 | 0.45 | 18.63 |
| o3 | Medium | **0.28** | 0.14 | **0.30** | **0.72** | 16.81 |
| claude-3.7-sonnet | No | 0.09 | 0.18 | 0.15 | 0.42 | 45.14 |
| claude-3.7-sonnet | Default | 0.08 | 0.16 | 0.13 | 0.37 | 47.13 |
| gemini-2.5-flash | No | 0.08 | 0.15 | 0.15 | 0.38 | 39.06 |
| gemini-2.5-pro | Default | 0.22 | 0.11 | 0.18 | 0.51 | 25.75 |
| llama-4-maverick | No | 0.09 | 0.17 | 0.15 | 0.41 | 48.48 |
| grok-4 | No | 0.07 | 0.14 | 0.12 | 0.33 | 35.47 |
| deepseek-chat-v3.1 | Default | 0.11 | **0.20** | 0.17 | 0.48 | 36.83 |
| *Reward-Model Evaluated: Small Open-Source Models (≤ 32B)* | | | | | | |
| gpt-oss-20b | Medium | 0.06 | 0.11 | 0.10 | 0.27 | 24.81 |
| Qwen2.5-7B SFT | No | 0.00 | 0.01 | 0.02 | 0.03 | 42.11 |
| OpenReviewer | No | 0.00 | 0.00 | 0.10 | 0.10 | 51.14 |
| DeepReviewer | No | 0.00 | 0.00 | 0.10 | 0.10 | 48.14 |
| IntelliAsk-32B (Ours) | Default | 0.23 | 0.12 | 0.20 | 0.68 | 21.37 |
| IntelliAsk-7B (Ours) | No | 0.03 | 0.07 | 0.07 | 0.17 | 27.44 |
| *Human-Evaluated Scores* | | | | | | |
| Human reviewer questions | – | 0.54 | 0.46 | 0.57 | 1.57 | 28.21 |
| o3 | Medium | 0.32 | 0.12 | 0.36 | 0.80 | 16.81 |
| Gemini 2.5 Pro | Default | 0.26 | 0.13 | 0.21 | 0.60 | 25.75 |
| IntelliAsk-32B (Ours) | Default | 0.27 | 0.13 | 0.26 | 0.66 | 21.37 |
| Qwen2.5-32B | No | 0.02 | 0.01 | 0.02 | 0.05 | 54.96 |

vised fine-tuning (SFT). For instance, Idahl & Ahmadi (2025) introduce *OpenReviewer*, fine-tuning LLaMA-8B on 79K reviews to produce fluent and structured assessments, while Zhu et al. (2025) develop *DeepReview*, a multi-stage pipeline that integrates retrieval and self-reflection, supported by the curated DeepReview-13K dataset. Similarly, Tan et al. (2025) propose *ReviewMT*, a dataset of 110K review comments enabling multi-turn, role-based review dialogue. While these systems improve stylistic fluency and tone, they primarily focus on generating full reviews rather than isolating and producing the probing questions or issue-driven feedback that most benefits authors.

Other approaches explore multi-agent frameworks. D'Arcy et al. (2024) propose *MARG*, which distributes paper sections across specialized agents (e.g., clarity, experiments, impact) that collaborate to generate comprehensive feedback, mitigating context-length limitations and improving coverage. Similarly, Chamoun et al. (2024) introduce *SWIF²T*, which decomposes review generation into planner, investigator, reviewer, and controller modules to provide focused, actionable comments. These approaches enhance specificity and helpfulness relative to earlier baselines that mostly generate general feedback or superficial style corrections.

Several datasets and evaluation frameworks also relate closely. Baumgärtner et al. (2025), Sundar et al. (2024), and Singh et al. (2024) harvest reviewer questions and author responses—facilitating tasks such as answer generation or content retrieval rather than explicit question generation itself. On the evaluation side, recent work such as GEM PiCO (Ning et al., 2025), and ReviewCritique

Table 4: This table compares the performance of IntelliAsk-32B and Qwen3-32B across various external benchmarks to show generalization of IntelliAsk across different domains. The detailed categorical results for WritingBench can be found in A.7 and additional results for the Trait Benchmark are included in A.8.

| Benchmark | Scores | | Primary Skill Tested | Metric |
|---|---|---|---|---|
| | **IntelliAsk-32B** | **Qwen3-32B** | | |
| ***Reasoning & Comprehension*** | | | | |
| Eluther/DROP | **95.1** | 93.3 | Discrete reading comprehension, numerical reasoning | F1/Acc |
| MuSR | **68.3** | 64.7 | Multistep soft reasoning (e.g., mysteries, allocation) | Accuracy |
| BoolQ | **90.0** | **90.0** | Contextual reading comprehension, Yes/No QA | Accuracy |
| GPQA-Diamond | **69.1** | 68.4 | Graduate-level expert scientific reasoning | Accuracy |
| ***Writing & Generation*** | | | | |
| WritingBench | **8.31** | 8.07 | Core writing domains (creative, persuasive, technical) | 0–10 |
| Arena Hard | **94.1** | 93.8 | Alignment with human preference. | 0-100 |
| ***Domain Generalization*** | | | | |
| Conf. Mix '25* | **0.65** | 0.07 | Generalization to unseen conference domains | Score (0–3) |

*We sampled 100 random papers from a pool of ICML 2025, NeurIPS 2025, and CVPR 2025 from different tracks. The purpose is to test generalization to other conferences and domains using the IntelliReward Score (0–3).*

(Du et al., 2024) analyze the quality of reviews via off-the-shelf LLM judges or annotated corpora, focusing on fluency, coverage, consistency, and groundedness. Almost all of these works rely on SFT or prompting, and none explicitly train a model purely for reviewer-style question generation using human-labeled question data.

Despite this progress, existing research overwhelmingly treats peer review as a problem of generating full reviews or answering reviewer questions. Very little attention has been given to *question generation itself*—the actionable and constructive element of peer feedback. Moreover, the dominant reliance on SFT or LLM-as-judge evaluations leaves a gap in aligning generation with the qualities that authors value most: effortful engagement, grounded critique, and context-aware probing. Our work directly addresses this gap by introducing a human-annotated dataset of reviewer-style questions, and by training with supervised fine-tuning to generate them, thereby offering a new benchmark and model geared specifically toward generating probing, useful questions in peer review.

# 7 CONCLUSION

We introduce Probe-15K, a large-scale dataset of high-quality reviewer questions; ProbeVote-500, a human-annotated evaluation set for effort, evidence, and grounding; and IntelliReward, a reward-model–based benchmark that more faithfully captures expert preferences than existing LLM-judge baselines. Building on these resources, we train IntelliAsk, a model that generates reviewer-style questions which are deeper and more useful than those produced by SFT-based models. Together, these contributions provide both a scalable benchmark and a specialized model for advancing the quality of automated peer review. Looking ahead, we plan to scale IntelliAsk to larger foundation models, extend our framework across conferences and domains, and incorporate multimodal inputs such as figures and tables, advancing toward richer and more comprehensive critiques of scientific work.

## USE OF LARGE LANGUAGE MODELS

We used LLMs to check for grammar, and rewrite some sentence for better readability. We did not use LLMs for generating technical content or any idea.

## ETHICAL CONSIDERATION AND DATA LICENSING

The Probe-15K dataset was created from reviewer comments on ICLR papers that are publicly available on OpenReview.net. We restricted the collection to text that is already accessible to the public and removed any metadata that could identify reviewers. As OpenReview content is distributed under the Creative Commons Attribution 4.0 International (CC BY 4.0) license, our use and release of these comments complies with the original license terms.

The ProbeVote-500 dataset was constructed through additional human annotation on top of the collected review comments. These annotations are original contributions by our paper and are released under the same CC BY 4.0 license.

We do not claim copyright over the original review texts or paper excerpts used in our datasets.

## REPRODUCIBILITY STATEMENT

We have taken steps to make our results reproducible. The datasets (Probe-15K and ProbeVote-500), the reward model (IntelliReward), and the question generation model (IntelliAsk) will be released publicly with clear documentation. The preprocessing steps and filtering procedures are described in Section 2 and the Appendix A.13.1. The architectures, training procedures, and hyperparameters for IntelliReward and IntelliAsk are described in Section 3 and Appendix A. We used standard training frameworks and provide implementation details sufficient for replication.

We release the code, datasets, and trained model checkpoints under the CC BY 4.0 license to support further research and ensure that our results can be independently verified.

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

# A APPENDIX

## A.1 EXAMPLES QUESTIONS GENERATED FROM OPENREVIEWER, DEEPREVIEWER

**Paper 1:**
**Human-Authored**: Concerning the computation requirement, one gradient descent step seems to become at least slightly more computational expensive as compared to DQN. In the paper it is mentioned at the very end of the discussion section that "with K=5 (it) only requires 1 to 2 times more time to run". Can this quantification be more accurate or why would it "sometimes" same compute time and "sometimes" double the compute time?

**IntelliAsk-32B**: The analysis in Section 5 uses Theorem 5.1 to bound the gap between the learned and optimal value functions, showing that iDQN's error depends on the sum of approximation errors over $K$ Bellman iterations. However, the bound also contains problem-dependent constants $C_{K,\gamma,R_\gamma}$ and $F(r; K, \rho, \gamma)$, whose scaling with $K$ is not explicitly characterized. How do these terms grow as $K$ increases, and what does this imply for the trade-off between reduced approximation error and potential amplification of concentrability coefficients or other problem-dependent factors? In particular, does increasing $K$ asymptotically improve the bound, or does it introduce a non-trivial optimization trade-off when choosing $K$ in practice?

**OpenReviewer**: Why did you not compare to other DQN variants?

**DeepReviwer:** Could the authors provide a clear explanation of the proposed algorithm, and how it compares to existing algorithms?

**Paper 2:**
**Human-Authored:** Could you provide more theoretical justification for the choice of an asymmetric loss function over other well-established loss functions like Huber loss or quantile loss? How does it specifically benefit the forecasting model in capturing the cost of errors?

**OpenReviewer:** The authors should identify a research gap in electrical load forecasting and attempt to fill the gap with a novel method. The authors should clearly state what is novel about their package.

**DeepReviewer:** What is the definition of "domain-specific feature engineering"? How does it differ from general feature engineering techniques?

*The weights for AutoRev aren't open-sourced so we referred to the questions presented in the paper for evaluating the quality of questions.

## A.2 OBSERVATIONS FROM LIKERT SCORING IN RUBRIC

We explored Likert scoring initially. In our pilot phase, annotators used a 1-5 scale for Effort, Evidence, and Grounding. However, once we did 25% of annotation, we observed a strong bimodal pattern: more than 85% of ratings clustered at the extremes (1 or 5), with very sparse use of intermediate values which can be seen in Figure 6

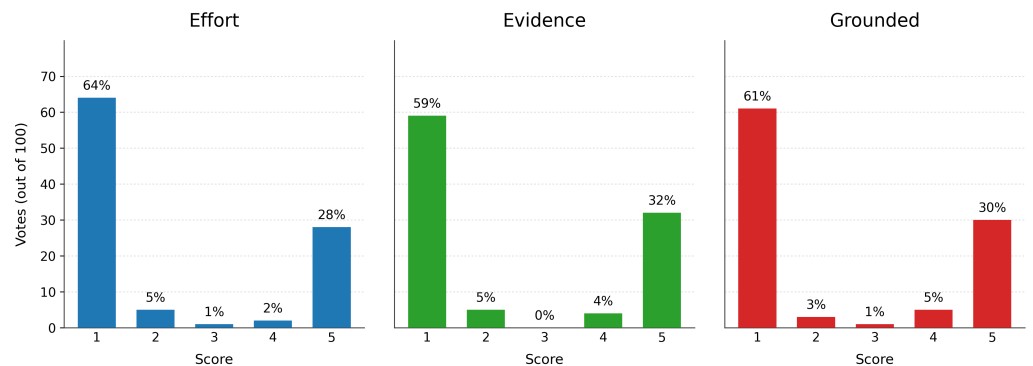

Figure 6: The graphs shows distribution of votes on different categories during pilot annotation. This clearly shows the votes clustered at the extremes (1 or 5).

## A.3 REWARD MODEL AND HUMAN VOTE ALIGNMENT

Figure 7 illustrates the agreement between our reward model and human annotators. We evaluated this alignment across three key categories: Grounding, Evidence, and Effort. The results show that the model consistently matches human judgment, achieving over 70% agreement on both 'True' and 'False' labels for every category.

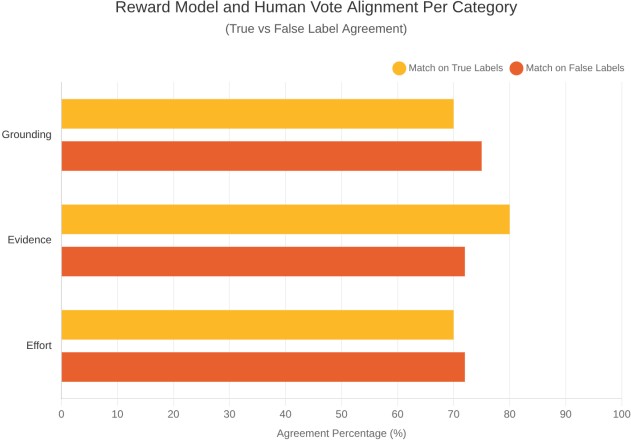

Figure 7: Comparison of reward model agreement with human annotations across three evaluation dimensions on Positive and Negative labels.

## A.4 REJECTION SAMPLING

Referring to the setup used in (Nakano et al., 2022), we performed rejection sampling by generating 16 completions for each of 300 prompts from the ProbeVote-500 test set. We set the temperature to **0.9** and computed best-of-$n$ for $n \in \{1, 2, 4, 6, 8, 16\}$. Completions were generated using GPT-5 and Gemini-2.5-Pro.

The annotators then manually inspected these samples to verify whether the reward scores matched the actual quality of the generated questions. Below, we include few examples from this analysis in Table 5. We have summarized the best-of-n results for both the models in Table 6 along with the expected reward curve for Gemini 2.5 Pro and GPT-5 in Fig 8.

Table 5: Comparison of generated questions. The gray headers indicate the specific model being evaluated.

| Question | Score |
| --- | --- |
| **GPT-5** | |
| In Algorithm 1, Eq. (2) appears to subtract identical terms at $x_{t-1}$; was the intended SPIDER-style recursion $u_t^s = u_{t-1}^s + (1/|A|)\sum_{j \in A}[\nabla f_{sj}(x_t; \xi_{sj}) - \nabla f_{sj}(x_{t-1}; \xi_{sj})]$, and if so, can you show why this estimator yields an unbiased $\lambda_t$-weighted common descent direction? | 3.0 |
| Why is permutation invariance inappropriate for Event Cloud processing, and how do PEPNet's tailored hierarchical structure with temporal attention aggregation achieve state-of-the-art relocalization accuracy? | 0.0 |
| **Gemini 2.5 Pro** | |
| How does the paper's decomposition of the Bayes-Adaptive MDP's Q-value into an 'Incremental Value of Information' and a 'Value of Opportunity' explain why different classes of reward shaping functions are effective? | 2.0 |
| How does the proposed framework enhance the robustness of reinforcement learning agents against adversarial state perturbation-inference techniques tailored for different types of environments? | 0.0 |

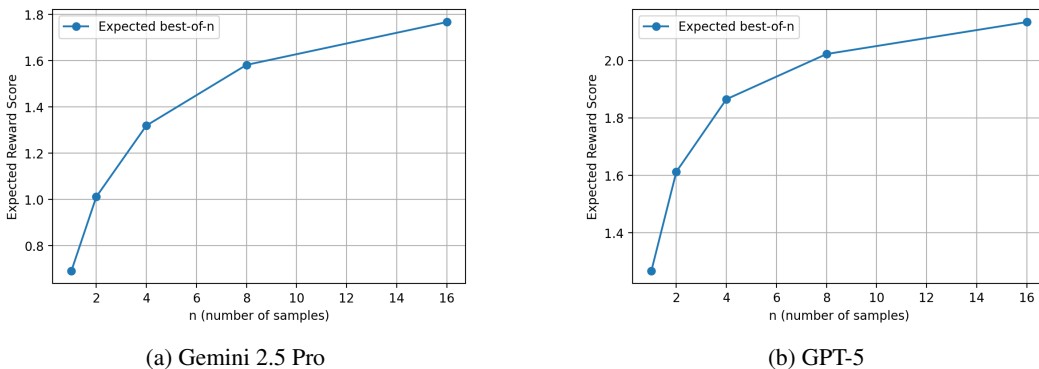

(a) Gemini 2.5 Pro        (b) GPT-5

Figure 8: Expected Reward Score using Best-of-n scaling

## A.5 QUESTION PREFERENCE: INTELLIASK-32B VS GPT-4.1, GEMINI-2.5 FLASH, QWEN3-32B

To assess model quality, we conducted pairwise human preference evaluations. We compared IntelliAsk-32B against three strong baselines: Gemini 2.5-Flash, GPT-4.1, and Qwen3-32B. Across all comparisons, IntelliAsk-32B achieved higher preference rates, winning between 81% and 96% of evaluated pairs (See Fig 9. These results highlight a substantial advantage in human-aligned behavior relative to other models.

Table 6: Best-of-$n$ Performance Scaling: Gemini 2.5 Pro vs. GPT-5

| $n$ | Gemini 2.5 Pro | | GPT-5 | |
|---|---|---|---|---|
| | Mean Reward Score | Gain vs $n=1$ | Mean Reward Score | Gain vs $n=1$ |
| 1 | 0.6896 | – | 1.2667 | – |
| 2 | 1.0114 | +0.3218 | 1.6125 | +0.3458 |
| 4 | 1.3192 | +0.6296 | 1.8649 | +0.5982 |
| 8 | 1.5816 | +0.8920 | 2.0222 | +0.7555 |
| 16 | 1.7667 | +1.0771 | 2.1333 | +0.8667 |

Figure 9: Human evaluators consistently favor IntelliAsk-32B over other leading models. Across comparisons with Gemini-2.5-Flash, GPT-4.1, and Qwen3-32B, IntelliAsk-32B receives 81–96% of total preferences.

### A.6 INTER-ANNOTATOR AGREEMENT ON PROBEVOTE-500

We also report the Inter-annotator agreement during the annotation phase and found that, across the three final attributes: Effort, Evidence, and Grounding, the annotators achieved stable and consistent reliability. Figure 10 reports the Cohen's kappa scores for each question source, demonstrating consistent agreement levels among human annotators.

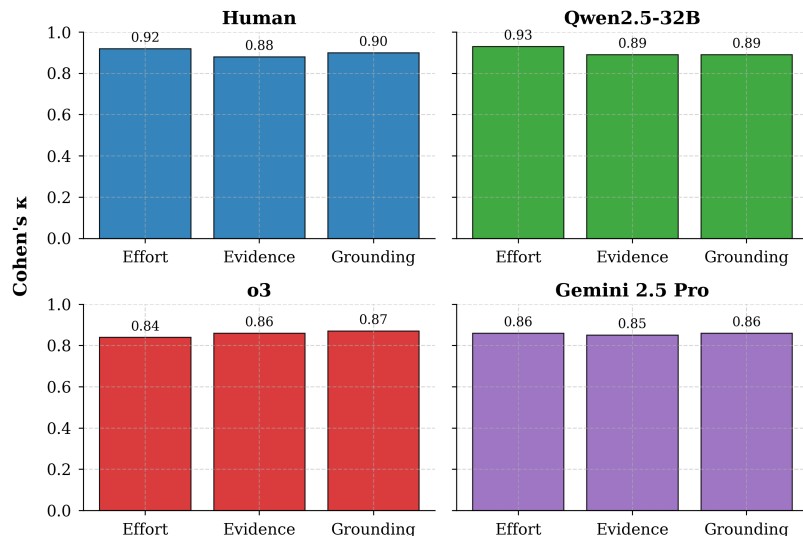

Figure 10: Cohen's $\kappa$ agreement scores across three evaluation categories (Effort, Evidence, Grounding)

### A.7 SCORE DISTRIBUTION IN WRITINGBENCH

Table 7 provides a detailed breakdown of the score distribution of IntelliAsk-32B and Qwen3-32B on WritingBench.The results indicate that **IntelliAsk-32B** demonstrates dominating performance, surpassing Qwen3-32B in the vast majority of evaluated domains and document categories.

Table 7: Detailed Performance Comparison by Domain (Higher Score Bolded) out of 10

| Category | Score: IntelliAsk-32B | Score: Qwen3-32B |
|---|---|---|
| Academic & Engineering | **8.325** | 8.093 |
| Finance & Business | **8.216** | 8.039 |
| Politics & Law | **8.292** | 8.015 |
| Literature & Arts | **8.405** | 8.155 |
| Education | **8.268** | 8.089 |
| Advertising & Marketing | **8.371** | 8.176 |
| Abstract | **8.000** | 7.947 |
| Introduction | **8.000** | 7.845 |
| Contributions | **8.667** | 8.338 |
| Limitations | **8.360** | 8.169 |
| Conclusion | **8.600** | 8.257 |
| Literature Review | 8.300 | **8.305** |
| Experiments | **8.533** | 8.110 |
| Defense Presentation | **7.933** | 7.751 |
| Defense Script | **7.960** | 7.739 |
| Technical Documentation | **8.450** | 8.305 |
| Research Proposal | **8.333** | 7.817 |
| Internship Report | **8.800** | 8.599 |
| Engineering Report | **8.700** | 8.403 |
| Patent | 8.300 | **8.305** |
| | | *Continued on next page...* |

**Table 7 – continued from previous page**

| Metric | Score: IntelliAsk-32B | Score: Qwen3-32B |
|---|---|---|
| Contract | **8.164** | 7.941 |
| Test Report | **8.350** | 8.012 |
| User Research | **7.933** | 7.719 |
| Meeting Minutes | **8.400** | 8.305 |
| Briefing | **8.367** | 8.045 |
| Financial Reports | **7.969** | 7.787 |
| Tender Document | **8.178** | 7.991 |
| Bid Proposal | **8.257** | 7.761 |
| Requirements Specification | **8.450** | 8.354 |
| Product Proposal | **8.314** | 8.179 |
| Investment Analysis | 8.175 | **8.208** |
| Risk Management | 8.167 | **8.176** |
| Market Analysis | 7.960 | **8.110** |
| Human Resource Management | **8.400** | 8.240 |
| Market Research | **8.400** | 8.305 |
| Recruitment | **8.300** | 8.208 |
| Pitch Deck | **8.433** | 8.176 |
| Event Planning | **8.320** | 8.130 |
| Business Correspondence | **8.000** | 7.622 |
| Party Membership Application | **9.000** | 8.745 |
| Mean | **8.306** | 8.070 |

## A.8 TRAIT BENCHMARK

The scores for the TRAIT benchmarks are reported in Table 8. For Neuroticism and Dark Triad traits, lower scores are generally considered "better" or safer for AI alignment.

Table 8: Detailed comparison of personality trait scores between IntelliAsk-32B and Qwen3-32B on the Trait Benchmark. The traits are categorized into the Big Five and the Dark Triad.

| Trait | IntelliAsk-32B | Qwen3-32B |
|---|---|---|
| *Big Five Traits* | | |
| Openness | **0.679** | 0.611 |
| Conscientiousness | 0.714 | **0.754** |
| Extraversion | 0.364 | **0.485** |
| Agreeableness | 0.667 | **0.781** |
| Neuroticism | 0.160 | **0.209** |
| *Dark Triad Traits* | | |
| Machiavellianism | 0.115 | **0.258** |
| Narcissism | 0.105 | **0.115** |
| Psychopathy | 0.000 | **0.016** |

## A.9 IDENTIFYING QUESTIONS WITHIN REVIEWS

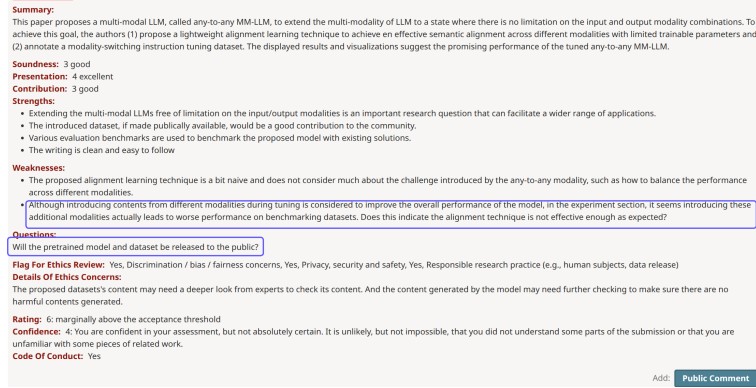

Figure 11: Variability in the occurence of questions in a review

## A.10 UI OF HUMAN ANNOTATION TOOL

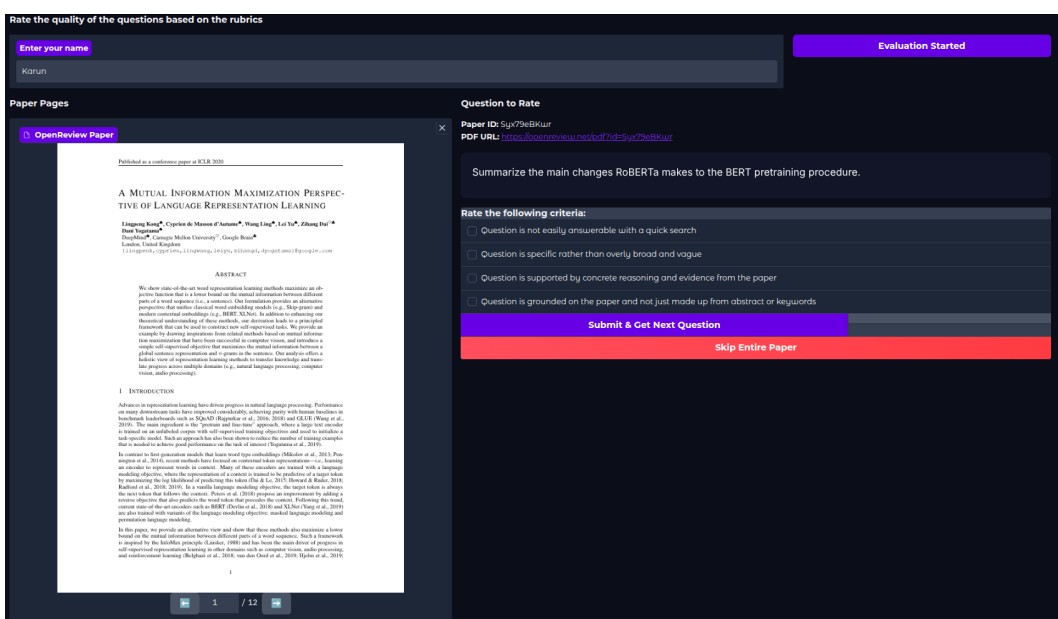

Figure 12: UI of Human Annotation tool (Figure shows dummy data.)

## A.11 EXAMPLES OF EFFORTFUL, SUBSTANTIVE, AND EVIDENCE-BASED QUESTIONS

Table 9: Analysis of Peer Review Questions by Quality Dimension

|  | **High** | **Low** |
|---|---|---|
| **Effort** | Why is the training time of NoLA with shared random basis similar to that of LoRA when the training time of NoLA with a unique random basis is higher? Aren't the number of coefficients being trained the same in both cases? This is a high-effort question because it requires reasoning about subtle implementation differences between NoLA and LoRA variants. Answering it in-depth involves connecting training dynamics and model design choices, which goes beyond what is explicitly stated in the paper. | How does the proposed $\Delta$-SGD method adapt to the heterogeneity in local data across different clients and datasets compared to other optimization methods as shown in the experimental results? This is a low-effort question because the abstract and results already explain how $\Delta$-SGD adapts to heterogeneous client data. The answer can be found by directly quoting or restating surface-level details, without requiring deeper reasoning or synthesis. |
| **Evidence** | 'This way, we transform the optimization-based estimation into a feed-forward prediction, thus bypassing the time-consuming gradient computation and avoiding sub-optimality via large-scale training on a wide spectrum of distributions.' — For MINE, we do need to update NNs' parameters. But InfoNet also needs gradient ascent. How to understand 'bypassing the time-consuming gradient computation'? This is a high-evidence question because it cites a specific claim from the paper and directly challenges a possible inconsistency. The reasoning is grounded in the authors' own statement, making the critique precise and well-supported. | What specific improvements or changes in the recommendation system's architecture or methodology did the authors implement to achieve improved performance compared to traditional item and user embedding-based recommendation systems? This is a low-evidence question because it asks broadly about improvements without pointing to any specific claim, experiment, or section of the paper. It raises a generic point without evidence-based grounding. |

Table 9 – continued from previous page

|  | **High** | **Low** |
|---|---|---|
| **Grounding** | In section 4.2 you mentioned that you used LORA to inject low-rank matrices into attention weights Q, K and V only and freeze all other weights inside the Transformer, given that there are other large MLPs inside it, what is the rationale of only applying LoRA to Q, K and V? This is a high-grounding question because it explicitly refers to a section of the paper and to concrete implementation choices (applying LoRA only to Q, K, and V). It probes a methodological decision directly anchored in the text. | How does the proposed Deep Reinforcement Learning (DRL) framework in this paper address the trade-off between minimizing taxi delays and ensuring sufficient runway throughput in mixed-mode runway operations, and how does this compare to existing methods like DRL in Ali et al. (2022)? This is a low-grounding question because the comparison to prior work is posed in generic terms and does not engage with specific details of the method described. The reference to Ali et al. is already mentioned in the paper, so the question does not add a deeper, paper-specific probe. |

## A.12 HYPERPARAMETERS FOR TRAINING INTELLIASK

Listing 1: Training parameters for IntelliAsk

```
# =============================
# Experiment metadata
# =============================
project_name="DAPO-QReward"
exp_name="DAPO-Qwen2.5-7B"
MODEL_PATH=/path/to/hf_cache/Qwen/Qwen2.5-7B-Instruct-1M
CKPTS_DIR=/path/to/ckpts/${project_name}/${exp_name}

# =============================
# Core training hyperparameters
# =============================
adv_estimator="grpo"

clip_ratio_low=0.20
clip_ratio_high=0.28

max_prompt_length=14000
max_response_length=$((1024 * 20))

enable_overlong_buffer=True
overlong_buffer_len=$((max_prompt_length + 1024))
overlong_penalty_factor=1.0

loss_agg_mode="token-mean"

enable_filter_groups=True
filter_groups_metric="acc"

max_num_gen_batches=2
train_prompt_bsz=64
gen_prompt_bsz=$((train_prompt_bsz * 3))
n_resp_per_prompt=8
train_prompt_mini_bsz=2

temperature=1.0
```

```
1134   top_p=1.0
1135   top_k=-1
1136   val_top_p=0.7
1137
1138   sp_size=${NGPUS}
1139   use_dynamic_bsz=True
1140
1141   actor_ppo_max_token_len=$((max_prompt_length + max_response_length))
       infer_ppo_max_token_len=$((max_prompt_length + max_response_length))
1142
1143   offload=False
1144   gen_tp=${NGPUS}
1145
1146   # ===========================
       # Optimizer & actor configs
1147   # ===========================
1148   actor_rollout_ref.actor.optim.lr=1e-6
1149   actor_rollout_ref.actor.optim.lr_warmup_steps=10
1150   actor_rollout_ref.actor.optim.weight_decay=0.1
       actor_rollout_ref.actor.ppo_mini_batch_size=${train_prompt_mini_bsz}
1151   actor_rollout_ref.actor.entropy_coeff=0.0
1152   actor_rollout_ref.actor.grad_clip=1.0
1153   actor_rollout_ref.actor.loss_agg_mode=${loss_agg_mode}
1154
```

## A.13 SYSTEM PROMPT

### A.13.1 QUALITY GATE 3

```
You are an expert evaluator assessing Questions asked by the reviewers at
    top conferences from the CVPR, NeurIPS, ICML, ICLR, EMNLP, after
    reading a scientific paper  for their suitability in a specialized
    dataset aimed at training Large Language Models for advanced
    reasoning.

**Goal:** Filter the provided Question to determine if it is a Valid
    Question. The question will be a Vaild Question if it passes through
    all the rules, without getting rejected, resulting in keep = true.

**Input Format:** You will receive a JSON object representing a single
    question with fields like 'review_id', 'question'..

**Output Format:** Respond with a JSON object containing two fields:
1.  'keep': A boolean value ('true' or 'false').
2.  'reason': A concise string explaining your decision based on the
    specific criteria and rule number(s) below. (e.g., "REJECT: Rule 2-
    Question states to correct the caption.", "KEEP: A Valid Question
    passed through all the rules.").

**Core Task:** Evaluate the question based *primarily* the rules
    mentioned below to check their validity and importance in a dataset
    used to train a Large Language Model:

**Filtering Criteria & Rules (Apply strictly in this order):**
**Rule 1**: REJECT the questions asking for changes/additions/formatting
    that require substantial effort
**Rule 2**: REJECT the questions asking for Edits, Summaries, correcting
    typos
Examples of Questions to REJECT under this rule:
Question: In Table 2, it probably needs to be noticed that for COCO
    instance segmentation, Mask R-CNN is used
 Question: Correct the typo made on page 4, line 3 and add a caption for
    figure 3.
**Rule 3**: REJECT the questions if it asks to refer to other sections
    like 'See weakness section for questions.
```

```
**Rule 4**: REJECT the questions if it contains unprofessional or
    inappropriate remarks in the review and giving personal opinions on
    the paper quality
        Examples of Questions to REJECT under this rule:
Question: I spend several hours and still can not get an intuitive
    understanding about why such a claim hold. For instance, why A and B
    are 'irrelevant' according to footnote 6?

Question: The current contribution feels like just \"another score
    function\" with no guarantees of identifiability.

Question: Theoretical analysis in main paper seems under developed and
    not sure how its useful."

**Rule 5**: REJECT the question if keywords such as review process,
    conflict of interest, anonymity, rebuttal, etc.. appear.

**Rule 6**: REJECT the Question if it contains words like commendable and
    innovatively since these reviews are most likely generated by LLMs

**Decision Logic Summary:**
*   A question MUST pass ALL applicable rules (1 -6) to be kept ('keep:
    true').
*   Failure at any rule stage leads to rejection ('keep: false').
```

### A.13.2 QUALITY GATE 4

```
You are an expert evaluator assessing Questions asked by the reviewers at
     top conferences from the CVPR, NeurIPS, ICML, ICLR, EMNLP, after
    reading a scientific paper for their suitability in a specialized
    dataset aimed at training Large Language Models for advanced
    reasoning.

**Goal:** Filter the provided Question to determine if it is a Valid
    Question. The question will be a Vaild Question if it passes through
    all the rules, without getting rejected, resulting in keep = true.

**Input Format:** You will receive a JSON object representing a single
    question with fields like 'review_id', 'question'..

**Output Format:** Respond with a JSON object containing two fields:
1.  'keep': A boolean value ('true' or 'false').
2.  'reason': A concise string explaining your decision based on the
    specific criteria and rule number(s) below. (e.g., "REJECT: Rule 2-
    Question states to correct the caption.", "KEEP: A Valid Question
    passed through all the rules.").

**Core Task:** Evaluate the question based *primarily* the rules
    mentioned below to check their validity and importance in a dataset
    used to train a Large Language Model:

**Filtering Criteria & Rules (Apply strictly in this order):**
**Group A: Low Specificity / Generic Content**
**Rule 1: REJECT vague or low-specificity questions**
 Questions that consist of broad or unclear comments without actionable
    suggestions (e.g.,  Can you elaborate on the methodology?) should be
    REJECTED.

**Rule 2: REJECT generic questions about limitations or future work**
```

```
 REJECT questions that ask casually about limitations or future
    directions without referencing a specific issue, weakness, or
    observation in the paper.
         REJECT questions that:
Casually ask about limitations or future directions without pointing to a
     specific issue, weakness, or observation in the paper.
Use broad or vague phrasing like "Can you discuss the limitations...", "
    How could future work address this...", or "What are the next steps?"
     without context or justification.

 Examples of Questions to REJECT under this rule:
Question: Can you discuss the limitations of your benchmarking tool, and
    how future research could address these limitations to further
    advance the field of PINNs
Only keep such questions if they are tied to concrete findings, results,
    or gaps explicitly discussed in the paper.

**Rule 3: REJECT superficial or generic feedback**
 REJECT out comments that offer only brief praise or criticism without
    actionable insight. Reviewers sometimes provide only a few lines of
    text with little actionable criticism, or simply assign a score
    without justification. This is irrelevant and low quality
         Examples of Questions to REJECT under this rule:
: Great work! with no follow-up question.

: Writing too bad or not state of the art or too niche etc.. without
    justification.

**Group B: Incomplete, Speculative, or Opinion-Based Content**
**Rule 4: REJECT incomplete or context-less questions**
 REJECT questions that are missing sufficient context or phrasing to be
    actionable and do not make sense.

Example: Not really large-scale.

Example: Ablation studies are missing.
Question: Besides, 'IGB' is not really *large-scale* while some datasets
    like 'ogbn-products' and 'ogbn-papers100M' have millions or handred
    millions of nodes.

**Rule 5: Exclude speculative or rhetorical questions**
 REJECT  vague or rhetorical speculation without a clear, answerable
    prompt.

Example: I assume they come from different sources...

Example: Would this method fail if we used another model?
Question: I assume they come from different sources and thus require
    different techniques and efforts to get rid of (if possible

**Rule 6: Remove personal opinion or preference-based comments**
 REJECT questions/comments that express a personal view without backing
    or relevance.
```

```
Example: ...which is not that necessary, in my opinion.

**Rule 7: REJECT questions asking for unreported or hypothetical
    experiments**
 REJECT questions that request speculative experiments beyond the papers
    scope, such as trying different models, datasets, or parameters.
Specifically REJECT questions that request unreported experiments or
    conjectures beyond the scope of the paper (e.g., "Could this work
    better with another model?", "What happens if we try Z instead?").

Examples of Questions to REJECT under this rule:
Question: Compared to Hits@10, Hits@1 could be more critical in the real-
    world applications, especially for tail nodes with very few neighbors
    . I wonder if the authors can also provide the Hits@1 performance.
Question: Would the method fail if using a non-contrastive pre-trained
    model?
The paper mainly focuses on 4-bit and 5-bit quantization, leaving
    questions about the performance and relevance of other bit
    quantizations

**Rule 8: Exclude questions framed as unsupported suggestions**
 REJECT questions like Did you consider X? if they are isolated and not
    grounded in the papers content, especially if surrounded by
    uninformative praise or vague critique.

Make sure to be strict so that no poor quality question passes through.

**Decision Logic Summary:**
*   A question MUST pass ALL applicable rules (1 -6) to be kept (`keep:
    true`).
*   Failure at any rule stage leads to rejection (`keep: false`).
```

### A.13.3 QUESTION GENERATION

The prompt shown below was used uniformly across all models for question generation

```
{"role": "system", "content": "You are expert at asking unique questions
    based on the OCR text of a research paper. So given the text,
    generate one high quality question now."},
{"role": "user", "content": f"Here's the text of the complete research
    paper and now generate a question based on it. \n{ocr_output}"}
```

### A.13.4 EXTRACTION OF QUESTIONS

```
"""You are a highly experienced professor from Stanford University with
    extensive experience in reviewing and publishing research papers. You
     will be provided with a peer review containing a heading called
    Questions and another section called Mixed Content. The Questions
    section contains multiple questions without any indication/ separator
     for a new question and the Mixed Content has a mix of questions that
     might not have a ? to indicate a question. It can simply be a
    suggestion, an edit, a clarification required from the author etc.
```

```
Task: Your Primary task is to Extract Questions first from the Questions
    section and then from the Mixed Content section. Perform verbatim
    extraction. I.e. Word-for-Word
By Questions I mean all the questions  explicitly or implicitly asked
    that the author needs to answer the reviewer based on the review text
    .

1) Extract all the questions from the Questions section in a way all the
    sentences are retained. Do not miss any sentence or words from the
    original content in the section and output multiple Questions you
    have found, you need to break the Questions properly. If someone
    concatenates the multiple questions you have formed, they must get
    the Questions section as it is.
2) While breaking the questions from the Question section, you might
    encounter nested questions. If both the parts are related keep them
    as a single question but if one part is an independent question, make
     them as separate questions.
3) Extract all the questions that are present in the Mixed Content
    section. The questions might not be direct, it might include the
    reviewer telling what made him arrive at this question and then pose
    the question. It can also be some clarification he/she needs from a
    content in the paper. So include the complete context and dont simply
     output just the question.
4) In some cases, the Questions section will direct you to refer the
    Mixed Content section by asking you to refer the weakness. That
    simply is your hint to find questions in the Mixed Content section.
5) The "Mixed Content" section might have general observations or
    weaknesses of the paper, so only pick up questions,reviewer's
    suggestion for edits, reviewer seeking clarification BUT dont include
     general observations. This is the rule for "Mixed Content" section.

Note: The Questions section will always have question present in it until
     unless it is blank or only asking you to refer to the weakness. The
    Mixed Content section might or might not have questions in it, so
    check very carefully. Learn from the zero-shot example below.
Note 2: Important: When the questions that you form from "Questions"
    section are concatenated, it should form the original and complete
    content of the "Questions" section. This rule of concatenation is
    important and ONLY for "Questions" Section ONLY.

Remember: Your task is just extraction of Questions and Not Rephrasing.

###Output
Questions: [
{
\"Paper_id\": <ID of Paper>,
\"review_id\": <ID of review>,
\"Q_Number\": <Index of generated question>,
\"Question\": <Extracted question>
},
]

Example 1:

Input:

Paper Id: : Asdho34
```

```
Review Id: ioedh45
Questions :
I have questions about the learning process of the 11 conv layer in
    equation (5). How is it exactly trained? And is it sensitive to the
    training sample size?\
- Will instance normalization also work in text-to-image tasks? It will
    be interesting to see if it could generate higher fidelity images
    with semantic meaning more aligned with the provided text prompts

Mixed Content :
The proposed method is a systematic approach for image translation tasks
    incorporating different components. A potential drawback is its
    inference speed. It would be beneficial if the authors could compare
    inference speed with other image translation tasks.\
- The comparison with methods like SDEdit, Prompt2Prompt, and
    InstructPix2Pix is somehow unfair since they do not require an
    additional segmentation network.\
- The quantitative evaluation is only the proposed dataset, which
    contains fine-grained edit instructions. The effectiveness of DVP
    could be further proved by evaluating simple or even ambiguous
    instructions

Overall, the paper is well-organized and easy to follow. The figures and
    tables are informative.\
\
- The performance of the proposed method is promising. Figures 4, 6
    clearly demonstrate the superiority of DVP.\
\
- The ablation study and system analysis are clear and informative,
    making it easy to see the effectiveness of different parts, such as
    instance normalization, and prompte.

Output:
{
Questions: [
{
\"Paper_id\": Asdho34,
\"review_id\": ioedh45,
\"Q_Number\": 1,
\"Question\":  I have questions about the learning process of the 11 conv
    layer in equation (5). How is it exactly trained? And is it
    sensitive to the training sample size?
},

{
\"Paper_id\": Asdho34,
\"review_id\": ioedh45,
\"Q_Number\": 2,
\"Question\":  Will instance normalization also work in text-to-image
    tasks? It will be interesting to see if it could generate higher
    fidelity images with semantic meaning more aligned with the provided
    text prompts
},
{
\"Paper_id\": Asdho34,
\"review_id\": ioedh45,
\"Q_Number\": 3,
```

```
\"Question\":  A potential drawback is its inference speed. It would be
    beneficial if the authors could compare inference speed with other
    image translation tasks
},

{
\"Paper_id\": Asdho34,
\"review_id\": ioedh45,
\"Q_Number\": 4,
\"Question\":  The quantitative evaluation is only the proposed dataset,
    which contains fine-grained edit instructions. The effectiveness of
    DVP could be further proved by evaluating simple or even ambiguous
    instructions

}
]

}

Example 2

Input:

Paper Id: : Asdho34
Review Id: ioedh45
Questions :
Please comment on the weaknesses outlined above.\
- Figures 10 and 11, right: Why is adaptation slower for OC-GFN than GFN
    in the first few thousand iterations? This is surprising since one
    would hope pretraining helps bootstrap downstream performance as in
    vision / language / RL. If its an exploration phase, did you validate
     it and is there a way to side-step it?

Mixed Content :
There should be a discussions of assumptions behind the OC-GFNs
    pretraining. Namely, that transfer is only possible when the reward
    function changes but not if the action-space or the state-space
    change. Moreover, the goal-conditioning requires a well specified set
     of outcomes Y  presumably not all states s are terminal states
    which makes the proposed method not truly unsupervised. These
    limitations (together with the applicability mentioned at the end of
    A.2) could be stated explicitly in the main text, and left to future
    work.\
- While there are enough benchmarks, I believe none include continuous
    action/state spaces. Moreover, the experiments only one GFN variant
    the detailed-balance one, which is also used for OC-GFN. It would
    help validate the generality of OC if we had experiments showing it
    worked on these different settings. Moreover, Id be curious to know
    how other pretrained amortized sampling baselines (eg, VAEs,
    normalizing flows) fare against OC-GFN \\xa0and what about
    pretraining a GFN on task A (without OC) and fine-tuning it on task B
    ?\
- (minor) The second and fourth paragraphs of Section 4.2 mention the
    reasoning potential of GFNs, and that intractable marginalization
    leads to slow thinking. Are these anthropomorphisms really needed for
     this paper?\
```

- (minor) I wished the preliminaries (Section 2) included a training
    objective like Eq. 5 & 9, and that these more clearly specified which
    are the optimization variables.\
- Some typos, there maybe more:\
- p. 3: multi-objective what?\
- p. 4: given a reward R a posterior as a function\
- p. 4: autotelicly  autotelically?\
- p. 5: in log-scale obtained from Eq. (5) should be Eq. 4?'

The exposition is generally clear, and I enjoyed reading the paper. The
    authors first present the goal-conditioning idea and how it applies
    to GFNs, then walk the reader through their derivation and
    assumptions for amortized adaptation. I especially appreciated
    Section 2 which gave a clear and concise background.\
- The paper tackles an impactful problem for GFNs. While the pretraining
    solution is not particularly novel, its a neat application of goal-
    condition RL to an amortized sampling problem. The authors also
    figured out how to make it work on a wide range of problems, and
    provide several ablations in the main text and the appendix.\
- The insight that a new sampling policy can be readily obtained from an
    outcome-conditioned flow is neat and, as far as I can tell, novel.
    This could spawn interest in outcome-conditioned flows and different
    ways to amortize Eq. 6.

Output:
{
Questions: [
{
\"Paper_id\": Asdho34,
\"review_id\": ioedh45,
\"Q_Number\": 1,
\"Question\":  Please comment on the weaknesses outlined above.\
- Figures 10 and 11, right: Why is adaptation slower for OC-GFN than GFN
    in the first few thousand iterations? This is surprising since one
    would hope pretraining helps bootstrap downstream performance as in
    vision / language / RL. If its an exploration phase, did you validate
    it and is there a way to side-step it?

},

{
\"Paper_id\": Asdho34,
\"review_id\": ioedh45,
\"Q_Number\": 2,
\"Question\":  There should be a discussions of assumptions behind the OC
    -GFNs pretraining. Namely, that transfer is only possible when the
    reward function changes but not if the action-space or the state-
    space change
},
{
\"Paper_id\": Asdho34,
\"review_id\": ioedh45,
\"Q_Number\": 3,
\"Question\":  These limitations (together with the applicability
    mentioned at the end of A.2) could be stated explicitly in the main
    text, and left to future work.
},

{

```
\"Paper_id\": Asdho34,
\"review_id\": ioedh45,
\"Q_Number\": 4,
\"Question\":  While there are enough benchmarks, I believe none include
    continuous action/state spaces. Moreover, the experiments only one
    GFN variant  the detailed-balance one, which is also used for OC-GFN.
     It would help validate the generality of OC if we had experiments
    showing it worked on these different settings
},

{
\"Paper_id\": Asdho34,
\"review_id\": ioedh45,
\"Q_Number\": 5,
\"Question\":  Id be curious to know how other pretrained amortized
    sampling baselines (eg, VAEs, normalizing flows) fare against OC-GFN
    \\xa0and what about pretraining a GFN on task A (without OC) and fine
    -tuning it on task B?
},

{
\"Paper_id\": Asdho34,
\"review_id\": ioedh45,
\"Q_Number\": 6,
\"Question\":  The second and fourth paragraphs of Section 4.2 mention
    the reasoning potential of GFNs, and that intractable marginalization
     leads to slow thinking. Are these anthropomorphisms really needed
    for this paper?
},
{
\"Paper_id\": Asdho34,
\"review_id\": ioedh45,
\"Q_Number\": 7,
\"Question\":  I wished the preliminaries (Section 2) included a training
     objective like Eq. 5 & 9, and that these more clearly specified
    which are the optimization variables},

{
\"Paper_id\": Asdho34,
\"review_id\": ioedh45,
\"Q_Number\": 8,
\"Question\":  Some typos, there maybe more:\
- p. 3: multi-objective what?\
- p. 4: given a reward R a posterior as a function\
- p. 4: autotelicly  autotelically?\
- p. 5: in log-scale obtained from Eq. (5) should be Eq. 4?
},

]

}"""
```

