# OpenReview forum: "IntelliAsk: Learning to Ask Critical Questions with Human-Aligned Rewards"
_ICLR.cc/2026/Conference — ICLR 2026 Conference Withdrawn Submission_

### Official Review · Reviewer_FrTU · 2025-10-18

**Soundness:** 2
**Presentation:** 2
**Contribution:** 2
**Rating:** 2
**Confidence:** 4

**Summary:**

The authors curate two datasets — Probe-15K (15.5k filtered reviewer questions) and ProbeVote-500 (expert-annotated with Effort, Evidence, and Grounding scores). They then train a reward model (IntelliReward) based on human preference data and use reinforcement learning (DAPO) to fine-tune an LLM (IntelliAsk) for question generation. Experimental results suggest that RL with the proposed reward model yields slightly more "human-like" questions than supervised fine-tuning (SFT).

**Strengths:**

- The paper is well-written and clearly describes the dataset construction and filtering pipeline.

- The release of Probe-15K and ProbeVote-500 may be useful for future studies on reviewer question generation or automated peer review.

**Weaknesses:**

-  The technical approach is essentially standard supervised or reinforcement learning with human-labeled data — a typical preference distillation setup. The work lacks novel modeling ideas or theoretical insights. Which means this work reads more like an engineering project for data cleaning and reward tuning than a research contribution that advances our understanding of question generation or alignment.


- The structure of the paper is confusing, with tables and figures difficult to locate, and important experimental details scattered across multiple sections or the appendix. The overall reading experience is quite poor.

- The evaluation is mainly an in-domain test, using the same question and paper.  Lack of proof of the generalization of the model.

**Questions:**

- How does IntelliAsk differ conceptually from standard RLHF or preference-based fine-tuning?

- Have you evaluated generalization on other venues (e.g., NeurIPS, CVPR) or non-ICLR review data?

---

> ### Author Response · Authors · 2025-11-21
> **Response to Reviewer FrTU (1/3): Our Contributions and proposed methods**
>
> > **The technical approach is essentially standard supervised or reinforcement learning with human-labeled data — a typical preference distillation setup. The work lacks novel modeling ideas or theoretical insights. Which means this work reads more like an engineering project for data cleaning and reward tuning than a research contribution that advances our understanding of question generation or alignment.**
>
> >**How does IntelliAsk differ conceptually from standard RLHF or preference-based fine-tuning?**
>
> Similar to our response(2/2) to Reviewer Vhs5:
> We don’t propose a new training method and our paper is focused on carefull construction of Dataset for improving question generation capabilities of LLM and a new Reward modeling architecture.
>
> Our contributions are not limited to applying existing methods. We introduce findings and design choices that, to our knowledge, have not been shown before. We design a new reward-modeling architecture that differs from the standard setup and achieves better performance.
>
> **(MLP vs. TransformerResidual)**
> To empirically validate our design, we extended our experiments to include a **Trainable Base Model** comparison. As shown in the table below, while unfreezing the base model improves the standard MLP baseline ($62\% \rightarrow 67\%$), it still fails to match our proposed architecture.
>
> Notably, our **Frozen Base + TransformerResidual** ($72\%$) outperforms the **Trainable Base + MLP** ($67\%$), confirming that the performance gain comes from our architectural design.
>
> | Base Model | Pooling | Head Type | Mean Accuracy (%) |
> | :--- | :--- | :--- | :--- |
> | **Standard Reward Model** | | | |
> | Frozen | None | MLP | 62 |
> | Frozen | Pool50 | MLP | 65 |
> | Trainable | None | MLP | 63 |
> | Trainable | Pool50 | MLP | 67 |
> | **IntelliReward (Ours)** | | | |
> | Frozen | None | Transf. Resid. | 69 |
> | Frozen | Pool50 | Transf. Resid. | **72** |
> | **Trainable** | **Pool50** | **Transf. Resid.** | **73** |
>
> *Note: Pool50 denotes mean pooling over the last 50 output tokens.*
>
>
> Our contributions also includes a dataset that requires extensive filtering based on several key observations and experiments. The voting data was created through many hours of human annotation, and no public dataset of similar quality exists. This dataset not only supports our analysis but also leads to models that improve their reasoning and generalize better across other benchmarks, which we attribute to the complexity and quality of the data.
>
> We release the full architecture, annotation data, and model weights so others can verify these results. The revised paper now includes additional ablations and experiments that further reinforce our claims, demonstrating that our reward models and dataset effectively support scaling to larger model sizes. We trained IntelliAsk-32B, which surpasses leading models across multiple benchmarks and evaluations, the same can be verified from results in table 3 and 4

---

> ### Author Response · Authors · 2025-11-21
> **Response to Reviewer FrTU (2/3): Generalization and Extended Benchmarks**
>
> > **The evaluation is mainly an in-domain test, using the same question and paper. Lack of proof of the generalization of the model.**
>
> >**Have you evaluated generalization on other venues (e.g., NeurIPS, CVPR) or non-ICLR review data?**
>
> We have extended evaluations on  **DROP**, **MuSR**, **BoolQ**, **GPQA-Diamond**, **WritingBench**, **TRAIT**, Papers from different domains of **ICML**, **CVPR**, **NeurIPS**, where IntelliAsk-32B consistently outperforms the Qwen3-32B (base model), which shows that our training also considerably improved the model on other tasks. We have reported these results below and the revised paper contains these results in Table 4
>
>
> **Table: This table compares the performance of IntelliAsk-32B and Qwen3-32B across various
> external benchmarks to show generalization of IntelliAsk across different domains. The detailed
> categorical results for WritingBench can be found in A.7 and additional results for the Trait Benchmark are included in A.8**
>
> | Benchmark | IntelliAsk-32B | Qwen3-32B | Metric |
> | :--- | :---: | :---: | :--- |
> | **_Reasoning & Comprehension_** | | | |
> | Eluther/DROP | **95.1** | 93.3 | F1/Acc |
> | MuSR | **68.3** | 64.7 | Accuracy |
> | BoolQ | **90.0** | **90.0** | Accuracy |
> | GPQA-Diamond | **69.1** | 68.4 | Accuracy |
> | **_Writing & Generation_** | | | |
> | WritingBench | **8.31** | 8.07 | 0–10 |
> | Arena Hard | **94.1** | 93.8 | 0-100 |
> | **_Domain Generalization_** | | | |
> | Conf. Mix '25* | **0.65** | 0.07 | Score (0–3) |
>
> In addition, we have reported the results of human evaluation on questions generated by IntelliAsk in Table 3 of the updated paper. Below are the results from the table.
>
> **Table: Human-Evaluated Scores on IntelliAsk-32B**
>
> | Model / Source | Reasoning | Effort | Evidence | Grounding | Total (0-3) | Coverage (%) |
> | :--- | :--- | :--- | :--- | :--- | :--- | :--- |
> | Human reviewer questions | -- | 0.54 | 0.46 | 0.57 | 1.57 | 28.21 |
> | o3 | Medium | 0.32 | 0.12 | 0.36 | 0.80 | 16.81 |
> | Gemini 2.5 Pro | Default | 0.26 | 0.13 | 0.21 | 0.60 | 25.75 |
> | IntelliAsk-32B (Ours) | Default | 0.27 | 0.13 | 0.26 | 0.66 | 21.37 |
> | Qwen2.5-32B | No | 0.02 | 0.01 | 0.02 | 0.05 | 54.96 |

---

> ### Author Response · Authors · 2025-11-21
> **Response to Reviewer FrTU (3/3): Changes in the updated paper**
>
> > **The structure of the paper is confusing, with tables and figures difficult to locate, and important experimental details scattered across multiple sections or the appendix. The overall reading experience is quite poor.**
>
> To make the content easier to navigate and consolidate related information, we have updated the paper with several structural and presentation improvements:
>
> - Trained **IntelliAsk-32B** and included **human evaluation results** and **benchmark evaluations** to illustrate generalization. For Human Eval See Table 3, for More Benchmarks See Table 4.
> - Extended evaluations on **DROP**, **MuSR**, **BoolQ**, **GPQA-Diamond**, **WritingBench**, **TRAIT**, and papers from **ICML**, **NeurIPS**, and **CVPR**. (See Table 4)
> - Updated the captions of **Table 1**, **Table 2**, and **Table 3** to improve clarity.
> - Added an extended **ablation study** in **Table 2**.
> - Included the **Reward Curve** for **Qwen 2.5-7B (SFT)** and **IntelliAsk** training in figure 5.
> - Added a visualization in **Appendix A.2** showing the clustering of extreme scores during Likert scoring.
> - Added a figure in **Appendix A.3** showing alignment between **IntelliReward** and human annotators.
> - Added **rejection sampling results** in **Appendix A.4** to illustrate how reward model scores distinguish good and bad questions.
> - Added the setup for **question-preference evaluation** between **IntelliAsk-32B** and other models in **Appendix A.5**.
> - Added the **inter-annotator agreement** figure in **Appendix A.6**.
> - Included the **UI setup** used by human annotators in **Appendix A.10**.

---

> ### Comment · Reviewer_FrTU · 2025-11-24
> **Response to the Authors**
>
> Thank you for your response. I think there's big room to improve the overall quality, especially in highlighting your main contribution. If your focus is the improved reward training technique built on previous work, I suggest rewriting the motivation to clearly position this as your core innovation, adding out-of-domain experiments to provide solid evidence of robustness, and demonstrating generalizability to show it's broadly applicable rather than just a narrow improvement.

---

> > ### Author Response · Authors · 2025-11-24
> >
> > Thanks you for your response. Our primary contributions are Probe-15K and ProbeVote-500 which were created with careful filtering stages based on our observations and several hours of human annotations. These datasets address a key limitation we observed, existing LLMs produce weak questions for review tasks as per our evaluations and hence generate poor quality reviews which don't help research in meaningful way. Using our datasets, we significantly improved the question-generation ability of open-source LLMs, surpassing several large closed-source models, as shown in Table 3 of the paper.
> >
> > We have drastically improved the quality of questions generated as compared to the existing research on peer review generation using our datasets and reward model.
> >
> > Our secondary contribution, IntelliReward, was introduced specifically to support training open-source LLMs for question generation, where current closed-source SOTA LLMs-as-judge have not worked well.
> >
> > We hope this asnwers your question and thank you once again for your suggestions.

---

### Official Review · Reviewer_Vhs5 · 2025-10-27

**Soundness:** 3
**Presentation:** 2
**Contribution:** 1
**Rating:** 2
**Confidence:** 3

**Summary:**

This paper creates resources for training models that ask questions about scientific articles as reviewers in the peer-review process would. They create a dataset of articles and associated review questions by filtering data from ICLR 2024. They define evaluation criteria for reward quality and use human annotators to compare the quality of LLM-generated questions and human-written questions. Next, they train a reward model to automate question scoring. They show that performing SFT is insufficient for generating high-quality reviews and that RL with their reward model can do better.

**Strengths:**

- The paper addresses an important problem of generating high-quality questions about scientific articles which is a part of understanding these articles and using models to accelerate science
- The authors are careful with their curation of the review question dataset, filtering for only high quality questions and performing deduplication.
- The work highlights an important property of SFT of primarily learning to mimic style and being insufficient for performing well at a desired but indirectly specified task (in this case, generating high-quality review questions)
- The reward model trained for assessing question quality is substantially stronger than strong baselines.

**Weaknesses:**

- From what I saw, IntelliAsk is only evaluated by the reward model IntelliReward. From what I understand, IntelliReward also provides the signal for training IntelliAsk. So, it seems possible that IntelliAsk is just reward hacking IntelliReward rather than actually producing better questions. Why isn’t there a human-evaluation for IntelliAsk?
- This paper largely applies existing methods to a novel dataset/task, so its contributions seem limited. I’m not sure what to take away from this paper besides the fact that generating review questions cannot be done well by current LLMs and SFT is insufficient for substantially improving at this task. A lot of the content of the paper is in describing the design details of the pipeline for creating the benchmark and not on insights.

**Questions:**

NA

---

> ### Author Response · Authors · 2025-11-21
> **Response to Reviewer Vhs5 (1/2): Human Evaluation and Benchmarking of IntelliAsk**
>
> We thank you for taking time and reviewing our paper in such short time. Below we address your concern and add extra human eval and benchmarks.
>
> > **From what I saw, IntelliAsk is only evaluated by the reward model IntelliReward. From what I understand, IntelliReward also provides the signal for training IntelliAsk. So, it seems possible that IntelliAsk is just reward hacking IntelliReward rather than actually producing better questions. Why isn’t there a human-evaluation for IntelliAsk?**
>
> We added human evals on our recently trained model IntelliAsk-32B and show that there was no reward hacking and also evaluated on more benchmarks.
>
> 1. Reward Hacking Analysis:  We added a human evaluation for IntelliAsk-32B and confirmed that its strong results are not caused by reward hacking. During training, we added N-gram repetition penalty, length based penalty to ensure the outputs are diverse and we also continued to monitor the output of the model throughout training to ensure the outputs are improving and not just exploiting the reward model. We have updated Table 3 in paper with below results.
>
> **Table: Human-Evaluated Scores**
>
> | Model / Source | Reasoning | Effort | Evidence | Grounding | Total (0-3) | Coverage (%) |
> | :--- | :--- | :--- | :--- | :--- | :--- | :--- |
> | Human reviewer questions | -- | 0.54 | 0.46 | 0.57 | 1.57 | 28.21 |
> | o3 | Medium | 0.32 | 0.12 | 0.36 | 0.80 | 16.81 |
> | Gemini 2.5 Pro | Default | 0.26 | 0.13 | 0.21 | 0.60 | 25.75 |
> | IntelliAsk-32B (Ours) | Default | 0.27 | 0.13 | 0.26 | 0.66 | 21.37 |
> | Qwen2.5-32B | No | 0.02 | 0.01 | 0.02 | 0.05 | 54.96 |
>
> 2. Benchmarks: We extended our evals to include **DROP**, **MuSR**, **BoolQ**, **GPQA-Diamond**, **WritingBench**, Papers from **ICML**, **NeurIPS**, **CVPR** 2025 where IntelliAsk-32B consistently outperforms the Base model Qwen3-32B, which shows that our training also considerably improved the model on other tasks like writing,  reasoning and domain generalization.
>
>
>
> **Table: This table compares the performance of IntelliAsk-32B and Qwen3-32B across various
> external benchmarks to show generalization of IntelliAsk across different domains. The detailed
> categorical results for WritingBench can be found in A.7 and additional results for the Trait Benchmark are included in A.8**
>
> | Benchmark | IntelliAsk-32B | Qwen3-32B | Metric |
> | :--- | :---: | :---: | :--- |
> | **_Reasoning & Comprehension_** | | | |
> | Eluther/DROP | **95.1** | 93.3 | F1/Acc |
> | MuSR | **68.3** | 64.7 | Accuracy |
> | BoolQ | **90.0** | **90.0** | Accuracy |
> | GPQA-Diamond | **69.1** | 68.4 | Accuracy |
> | **_Writing & Generation_** | | | |
> | WritingBench | **8.31** | 8.07 | 0–10 |
> | Arena Hard | **94.1** | 93.8 | 0-100 |
> | **_Domain Generalization_** | | | |
> | Conf. Mix '25 | **0.65** | 0.07 | Score (0–3) |
>
> 3. Human Eval: We expanded our human preference study and included preference annotation to show that IntelliAsk-32B questions are more preferred than output from other leading models. Figure 9 in Section A.5 of paper shows humans largely preferred question generated by IntelliAsk-32B that other leading models (Gemini 2.5 Flash, GPT-4.1, Qwen3-32B)
>
>
> **Table: Human evaluators consistently favor IntelliAsk-32B over other leading models. Across
> comparisons with Gemini-2.5-Flash, GPT-4.1, and Qwen3-32B, IntelliAsk-32B receives 81–96% of
> total preferences.**
>
> | Competitor Model | IntelliAsk-32B Preference | Competitor Preference |
> | :--- | :---: | :---: |
> | **Gemini2.5-flash** | 81% | 19% |
> | **GPT-4.1** | 85% | 15% |
> | **Qwen3-32B** | 96% | 4% |

---

> ### Author Response · Authors · 2025-11-21
> **Response to Reviewer Vhs5 (2/2): Contributions**
>
> > **This paper largely applies existing methods to a novel dataset/task, so its contributions seem limited. I’m not sure what to take away from this paper besides the fact that generating review questions cannot be done well by current LLMs and SFT is insufficient for substantially improving at this task. A lot of the content of the paper is in describing the design details of the pipeline for creating the benchmark and not on insights.**
>
> Our contributions are not limited to applying existing methods. We introduce findings and design choices that, to our knowledge, have not been shown before. We design a new reward-modeling architecture that differs from the standard setup and achieves better performance.
>
> **(MLP vs. TransformerResidual)**
> To empirically validate our design, we extended our experiments to include a **Trainable Base Model** comparison. As shown in the table below, while unfreezing the base model improves the standard MLP baseline ($62\% \rightarrow 67\%$), it still fails to match our proposed architecture.
>
> Notably, our **Frozen Base + TransformerResidual** ($72\%$) outperforms the **Trainable Base + MLP** ($67\%$), confirming that the performance gain comes from our architectural design.
>
> | Base Model | Pooling | Head Type | Mean Accuracy (%) |
> | :--- | :--- | :--- | :--- |
> | **Standard Reward Model** | | | |
> | Frozen | None | MLP | 62 |
> | Frozen | Pool50 | MLP | 65 |
> | Trainable | None | MLP | 63 |
> | Trainable | Pool50 | MLP | 67 |
> | **IntelliReward (Ours)** | | | |
> | Frozen | None | Transf. Resid. | 69 |
> | Frozen | Pool50 | Transf. Resid. | **72** |
> | **Trainable** | **Pool50** | **Transf. Resid.** | **73** |
>
> *Note: Pool50 denotes mean pooling over the last 50 output tokens.*
>
>
> Our contributions also includes a dataset that requires extensive filtering based on several key observations and experiments. The voting data was created through many hours of human annotation, and no public dataset of similar quality exists. This dataset not only supports our analysis but also leads to models that improve their reasoning and generalize better across other benchmarks, which we attribute to the complexity and quality of the data.
>
> We release the full architecture, annotation data, and model weights so others can verify these results. The revised paper now includes additional ablations and experiments that further reinforce our claims, demonstrating that our reward models and dataset effectively support scaling to larger model sizes. We trained IntelliAsk-32B, which surpasses leading models across multiple benchmarks and evaluations, the same can be verified from results in table 3 and 4

---

### Official Review · Reviewer_7t4H · 2025-10-29

**Soundness:** 3
**Presentation:** 3
**Contribution:** 2
**Rating:** 6
**Confidence:** 4

**Summary:**

This paper studies the problem of generating critical review questions for peer review using LLMs. The authors conduct a study in which they compile a dataset of 15,000 reviewer questions extracted from ICLR 2024 papers through a multi-stage filtering process designed to identify high-quality review questions. They also compare these questions with questions generated using powerful reasoning models and find a substantial quality gap between human and LLM-generated questions. To address this gap, the authors propose a reward model, called IntelliReward, that evaluates the quality of review questions along three dimensions: effort, evidence, and grounding. Then, they use this reward model to train a question generation model (IntelliAsk) using reinforcement learning and show that the questions generated by IntelliAsk outperform those generated by a model trained solely through supervised fine-tuning.

**Strengths:**

- The paper is easy to follow and well-written.
- The proposed datasets, Probe-15K and ProbeVote-500, are valuable resources for advancing research in high-quality question generation and evaluation.
- The fact that SFT does not help much with creating better question generation models is an important problem studied in this paper, and their proposed solution for training a reward model is interesting.

**Weaknesses:**

- The design of IntelliReward is not well-motivated. How did the authors come up with the frozen causal LLM + TransformerResidualHead idea?
- Some statistics about the ProveVote-500 are missing. It’s not clear what the score distribution is for the 3 dimensions (effort, evidence, and grounding) in this dataset. Knowing these statistics can help understand Table 1 better.
- The rubric designed to assess question quality is not well motivated. Also, using a 0/1 score for a question seems to be a simplistic assumption.

**Questions:**

- Why do you define keyword coverage like that? Can you provide more analysis on why this metric is a good representative of the coverage of a question?
- Can you provide an analysis of how much the IntelliReward score aligns with Human Evaluated scores for question quality evaluation?
- Why does it make sense to evaluate IntelliAsk using IntelliReward?
- Why is IntelliAsk failing in the coverage metric compared to SFT, OpenReviewer, and DeepReviewer (table 3)?

---

> ### Author Response · Authors · 2025-11-21
> **Response to Reviewer 7t4H (1/7):  IntelliReward Motivation**
>
> We sincerely appreciate the reviewer’s time and thoughtful feedback. We address each of the raised points below, and we hope that our responses help clarify the concerns.
>
> > **1. The design of IntelliReward is not well-motivated. How did the authors come up with the frozen causal LLM + TransformerResidualHead idea?**
>
> We proposed a new reward model design which is not conventional and surpasses the old design, scales better and is more efficient.
> We started with a conventional setup that passed hidden states through an MLP layer, but after training the average accuracy concentrated at around 66%. We also observed that allowing the model to reason before producing an answer and pooling the hidden states from the last 50 tokens and TransformerResidualHead on top gave the best results. Beyond 50 tokens, the accuracy again concentrated, with a mean of 72%.
>
> We have updated Table 2 in the paper with ablations that are reported below.
>
>
> **Table: Ablation study on reward model architecture. HS = Hidden State. Pool-k = mean pooling over last k output tokens.**
>
> | HS     | Pooling  | Head            | Effort | Evidence | Grounding | Mean |
> |--------|----------|-----------------|--------|----------|-----------|------|
> | Output | None     | Transf. Resid.  | 68     | 68       | 70        | 69   |
> | Output | Pool50   | Transf. Resid.  | **70** | **76**   | **70**    | **72** |
> | Output | Pool128  | Transf. Resid.  | 69     | 77       | 67        | 71   |

---

> ### Author Response · Authors · 2025-11-21
> **Response to Reviewer 7t4H (2/7):  Distribution for Effort, Evidence and Grounding in ProbeVote-500**
>
> >**2. Some statistics about the ProveVote-500 are missing. It’s not clear what the score distribution is for the 3 dimensions (effort, evidence, and grounding) in this dataset. Knowing these statistics can help understand Table 1 better.**
>
> We have revised the paper and added a visualization (Figure 3) that illustrates the table below.
>
> **Table: The table shows the distribution of percentage of votes on Effort, Evidence and Factual metrics for
> various sources of questions**
>
> | Source                   | High Effort (%) | Evidence (%) | Factual (%) |
> |--------------------------|------------------|--------------|-------------|
> | Qwen-2.5 32B             | 2                | 1            | 2           |
> | Gemini 2.5 Pro           | 26               | 13           | 21          |
> | o3                       | 32               | 12           | 36          |
> | Human Authored Questions | 54           | 46       | 58      |

---

> ### Author Response · Authors · 2025-11-21
> **Response to Reviewer 7t4H (3/7): Motivation for our Rubric**
>
> >**3. The rubric designed to assess question quality is not well motivated. Also, using a 0/1 score for a question seems to be a simplistic assumption.**
>
> Our rubric was developed through multiple rounds of pilot analysis in which we compared questions from different sources and examined which attributes most reliably distinguished strong, well-reasoned questions. We initially explored a larger set of metrics, but many were highly correlated or difficult for annotators to separate in practice. After iterative testing, we narrowed down to only four factors- Effort, Evidence, Grounding, and Specificity.
>
> During pilot annotation, **‘Specificity’** (which measured whether a question is specific rather than overly broad and vague) showed substantially lower inter-annotator agreement **(κ = 0.59)**,  so we removed it to maintain reliability and keep the final annotation set focused on attributes with higher agreement.Across the three final attributes (Effort, Evidence, and Grounding), annotators achieved stable and consistent agreement and we have reported their Cohen’s score in the table below.
>
> We also tested Likert scoring. As described in our response(1/6: Why not Likert Scores) to **Reviewer Xtfe** , annotators used a 1-5 scale for Effort, Evidence, and Grounding. However, once we did 25% of annotation, we observed a strong bimodal pattern: more than 85% of ratings clustered at the extremes (1 or 5), with very sparse use of intermediate values. We have updated the paper to include this visualization in Appendix A.2.
>
> **Table: The table shows distribution of votes on different categories during pilot annotation.
> This clearly shows the votes clustered at the extremes (1 or 5)**
>
> | Category   | Score 1 | Score 2  | Score 3 | Score 4  | Score 5 |
> |------------|-------------|-------------|-------------|-------------|-------------|
> | Effort     | 64%         | 5%          | 1%          | 2%          | 28%         |
> | Evidence   | 59%         | 5%          | 0%          | 4%          | 32%         |
> | Grounded   | 61%         | 3%          | 1%          | 5%          | 30%         |

---

> ### Author Response · Authors · 2025-11-21
> **Response to Reviewer 7t4H (4,5/7): Keyword Coverage & Alignment of IntelliReward and Human Annotators**
>
> >**4. Why do you define keyword coverage like that? Can you provide more analysis on why this metric is a good representative of the coverage of a question?**
>
> Keyword coverage is a simple diagnostic and not a new metric that we use to quantify a specific behavior we observed during inspection: Some models, especially Qwen, often form questions by reusing wording from the abstract or early paragraphs by adding words like “how” or “why” to them. To capture this, we measure the fraction of a question’s tokens that also appear on the first page. Standard overlap metrics such as Jaccard are not very informative here because the first page is much longer than the question, which pushes their values into very small number that do not reflect how much of the question is actually built from page-1 content.
>
> >**5. Can you provide an analysis of how much the IntelliReward score aligns with Human Evaluated scores for question quality evaluation?**
>
> The below table shows alignment/agreement of more than 70% on each binary label(True, False) across 3 metrics(Grounding, Evidence, Effort). We have added the corresponding figure in Appendix (Figure 7 Section A.3).
>
> | Category   | Match on True Labels (%) | Match on False Labels (%) |
> |------------|---------------------------|----------------------------|
> | Effort     | 70                        | 72                         |
> | Evidence   | 80                        | 72                         |
> | Grounding  | 70                        | 75                         |

---

> ### Author Response · Authors · 2025-11-21
> **Response to Reviewer 7t4H (6/7): Extended Eval & Benchmarks of IntelliAsk**
>
> >**6. Why does it make sense to evaluate IntelliAsk using IntelliReward?**
>
> To make sure IntelliAsk is not exploiting or “reward hacking” this model, we have trained IntelliAsk-32B and have reported the results of human evaluation on questions generated by IntelliAsk in Table 3 of the updated paper. Below are the results from the table.
>
> **Table: Human-Evaluated Scores**
>
> | Model / Source | Reasoning | Effort | Evidence | Grounding | Total (0-3) | Coverage (%) |
> | :--- | :--- | :--- | :--- | :--- | :--- | :--- |
> | Human reviewer questions | -- | 0.54 | 0.46 | 0.57 | 1.57 | 28.21 |
> | o3 | Medium | 0.32 | 0.12 | 0.36 | 0.80 | 16.81 |
> | Gemini 2.5 Pro | Default | 0.26 | 0.13 | 0.21 | 0.60 | 25.75 |
> | IntelliAsk-32B (Ours) | Default | 0.27 | 0.13 | 0.26 | 0.66 | 21.37 |
> | Qwen2.5-32B | No | 0.02 | 0.01 | 0.02 | 0.05 | 54.96 |
>
>
>
> In addition, we have done another round of human preference annotation of IntelliAsk-32B, from which it’s clearly evident that humans highly prefer question generated by IntelliAsk-32B than other models(Qwen3-32B, Gemini 2.5 Flash, GPT-4.1). We have updated the paper with the addition of below statistics in Appendix A.5.
>
>
> **Table: Human evaluators consistently favor IntelliAsk-32B over other leading models. Across
> comparisons with Gemini-2.5-Flash, GPT-4.1, and Qwen3-32B, IntelliAsk-32B receives 81–96% of
> total preferences.**
>
> | Competitor Model | IntelliAsk-32B Preference | Competitor Preference |
> | :--- | :---: | :---: |
> | **Gemini2.5-flash** | 81% | 19% |
> | **GPT-4.1** | 85% | 15% |
> | **Qwen3-32B** | 96% | 4% |
>
>
> We have extended evaluations on  **DROP**, **MuSR**, **BoolQ**, **GPQA-Diamond**, **WritingBench**, **TRAIT**, Papers from different domains of **ICML**, **CVPR**, **NeurIPS**, where IntelliAsk-32B consistently outperforms the Qwen3-32B (base model), which shows that our training also considerably improved the model on other tasks. We have reported these results below and the revised paper contains these results in Table 4
>
>
> **Table: This table compares the performance of IntelliAsk-32B and Qwen3-32B across various
> external benchmarks to show generalization of IntelliAsk across different domains. The detailed
> categorical results for WritingBench can be found in A.7 and additional results for the Trait Benchmark are included in A.8**
>
> | Benchmark | IntelliAsk-32B | Qwen3-32B | Metric |
> | :--- | :---: | :---: | :--- |
> | **_Reasoning & Comprehension_** | | | |
> | Eluther/DROP | **95.1** | 93.3 | F1/Acc |
> | MuSR | **68.3** | 64.7 | Accuracy |
> | BoolQ | **90.0** | **90.0** | Accuracy |
> | GPQA-Diamond | **69.1** | 68.4 | Accuracy |
> | **_Writing & Generation_** | | | |
> | WritingBench | **8.31** | 8.07 | 0–10 |
> | Arena Hard | **94.1** | 93.8 | 0-100 |
> | **_Domain Generalization_** | | | |
> | Conf. Mix '25 | **0.65** | 0.07 | Score (0–3) |

---

> ### Author Response · Authors · 2025-11-21
> **Response to Reviewer 7t4H (7/7): Clarification about Keyword Coverage**
>
> >**7 . Why is IntelliAsk failing in the coverage metric compared to SFT, OpenReviewer, and DeepReviewer (table 3)?**
>
> We have clarified the definition of the coverage metric in the revised version of the paper: Lower coverage indicates better behavior, since high coverage indicates that a question is formed primarily from the first page of the paper. Under this definition, IntelliAsk is not failing the metric,  its lower coverage score shows that it relies less on introductory text and draws from deeper sections of the paper.

---

### Official Review · Reviewer_Xtfe · 2025-11-02

**Soundness:** 2
**Presentation:** 2
**Contribution:** 2
**Rating:** 2
**Confidence:** 4

**Summary:**

This paper aims at generating better questions using language models for reviewing papers. It makes two contributions:
1. The authors release a dataset and trains a reward model for assessing review question qualities
2. To demonstrate the effectiveness of the reward model, they train another language model to generate questions regarding a paper, and shows it can achieve better performance compared to a baseline.

**Strengths:**

1. The paper addresses an interesting and important problem: the current language models often ask superficial questions and it’s interesting to study how to enable these LMs asking more in-depth and insightful questions, which is important for building automating scientific workflows.
2. The released dataset and resource can be helpful for the community, and the automatic dataset curation pipeline and cleaning code can be useful for other use cases.

**Weaknesses:**

1. While I like the motivation of this work, one major weakness of this paper is its **experimental design**. I have a lot of questions regarding the experimental details, and I think there are quite a few parts that are not rigorous and impact the soundness.
2. Also the presentation of the paper can be improved
    - For example, in section 4, there’s no pointer to the evaluation results of the model in table 3.
    - For the tables, please use consistent numerical styles, i.e., choosing either percentage or absolute numbers and use it in all tables, rather than using different styles in table 1 and 3.

**Questions:**

In the annotation process:
1. Why only using a binary score but not likert scores?
2. Also have you explored with another annotation method – for example, there can be multiple review questions for a paper. In that case, another approach is to have people label “preferences”,  i.e., contrasting two or more questions and picking the preferred ones.
3. What is the inter-annotator agreement in the annotation? I can imagine it’s relatively hard to achieve good agreement among people given this super challenging dataset

In the experiments design and evaluation
1. Sec 4 makes the claim that SFT may not be suitable for this task. However the evaluation seems to be less rigorous:
    - If the evaluation is correct, it can only prove that training on the 15k data may not be helpful, and it’s hard to make the general claim that SFT is less useful.
    - Also the evaluation seems to be relatively less rigorous – there’s only some qualitative studies on the annotation qualities and some approximated model-based evaluations.
2. How is the Intellireward evaluated? As far as I can tell, it seems the authors fine-tuned the LLM on the ProbeVote-500 dataset, and then conducted  evaluation on the (training) data? Then it seems natural that the model can achieve way better performance compared to the other baselines reported in table 1.
3. Also the comparison between the IntelliAsk model and the Qwen2.5-7B SFT model seem to be not fair as they are trained on completely different datasets. Why not directly conducting SFT on the “preferred questions” rated by IntelliAsk and compare the model performances?

---

> ### Author Response · Authors · 2025-11-21
> **Response to Reviewer Xtfe (1/6): Why not likert scores**
>
> We appreciate your time and feedback. We have addressed all your concerns below and we hope our responses clarify the issues.
>
> > **1. Why only using a binary score but not likert scores?**
>
> We explored Likert scoring initially. In our pilot phase, annotators used a 1-5 scale for Effort, Evidence, and Grounding. However, once we did around 25% of annotation, we observed a strong bimodal pattern: more than 85% of ratings clustered at the extremes (1 or 5), with very sparse use of intermediate values as evident from the table below.
>
> Annotators said they mostly used extreme scores because the questions were either grounded or not, effortful or not, and evidence-based or not. They explained that it was very hard and ambiguous to assign intermediate scores (2-4), especially for short questions where fine-grained distinctions are less consistent than in long paragraphs. We have updated the paper to include this visualization in Appendix A.2.
>
> | Category   | Score 1 | Score 2  | Score 3 | Score 4  | Score 5 |
> |------------|-------------|-------------|-------------|-------------|-------------|
> | Effort     | 64%         | 5%          | 1%          | 2%          | 28%         |
> | Evidence   | 59%         | 5%          | 0%          | 4%          | 32%         |
> | Grounded   | 61%         | 3%          | 1%          | 5%          | 30%         |

---

> ### Author Response · Authors · 2025-11-21
> **Response to Reviewer Xtfe (2/6): Other annotation methods**
>
> > **2. Also have you explored with another annotation method – for example, there can be multiple review questions for a paper. In that case, another approach is to have people label “preferences”, i.e., contrasting two or more questions and picking the preferred ones.**
>
> We considered preference-based annotation, but it did not align well with our setup. Our evaluation uses three independent binary attributes: **Effort**, **Evidence**, and **Grounding**, rather than a single continuous notion of quality. In pairwise settings, one question may be better on one attribute and worse on another, which makes a single “preference” ambiguous and would require three separate comparisons per pair. For short reviewer questions, annotators also found these attributes to be naturally present-or-absent, so binary labels produced clearer and more consistent judgments.
>
> We also aimed to keep the annotation focused on our main goal: distinguishing high-quality human questions from weaker model-generated ones. In practice, model-generated questions were consistently poor across all attributes, so preference voting added little value. While preference judgments among human questions were possible, those questions had already passed a rigorous multi-step filtering and were uniformly high quality, making pairwise annotation unnecessarily complex without meaningful benefit.

---

> ### Author Response · Authors · 2025-11-21
> **Response to Reviewer Xtfe (3/6):  Inter-annotator agreement in the annotation**
>
> > **3. What is the inter-annotator agreement in the annotation? I can imagine it’s relatively hard to achieve good agreement among people given this super challenging dataset**
>
> We measured agreement between annotators during the annotation phase. Across the three final attributes (Effort, Evidence, and Grounding), annotators achieved stable and consistent agreement. The table below shows the **Cohen's κ scores** measuring agreement among human annotators for each question source.
>
> Before finalizing the labeling protocol, we conducted multiple pilot rounds to resolve ambiguities in the rubric and ensure all annotators shared a consistent understanding of each attribute. We initially included a fourth attribute, **Specificity**, which measured whether a question is specific rather than overly broad and vague. This dimension showed higher variability (**κ = 0.59**), so we removed it to maintain reliability and keep the final annotation set focused on attributes with higher agreement.
>
> We have revised the paper accordingly and added the corresponding visualization in **Appendix A.6**. The table summarizes the agreement patterns across all sources.
>
>
>
> | Source        | Effort | Evidence | Grounding |
> |---------------|--------|----------|-----------|
> | **Human**     | 0.92   | 0.88     | 0.90      |
> | **Qwen2.5-32B** | 0.93 | 0.89     | 0.89      |
> | **o3**            | 0.84   | 0.86     | 0.87      |
> | **Gemini 2.5 Pro** | 0.86 | 0.85 | 0.86 |
>
>
> Table: Cohen’s $\kappa$ agreement scores across three evaluation categories (Effort, Evidence, Grounding)

---

> ### Author Response · Authors · 2025-11-21
> **Response to Reviewer Xtfe (4/6): Performance of SFT on our task**
>
> > **If the evaluation is correct, it can only prove that training on the 15k data may not be helpful, and it’s hard to make the general claim that SFT is less useful.**
>
> We want to clarify that we are not making a general claim that SFT is less useful. Our point is specific to this task: even after training Qwen2.5-7B-Instruct on 15k filtered, high-quality questions, the SFT reward curve remained flat as seen in our newly added **Figure 5 in Section 5.3** indicating no improvement on our rubric. In contrast, IntelliAsk trained with RLHF (using IntelliReward) shows a clear upward reward trend. The curves in the appendix illustrate this difference. Our conclusion is limited to this setting - SFT alone did not improve alignment on our rubric, while reward-based optimization did.

---

> ### Author Response · Authors · 2025-11-21
> **Response to Reviewer Xtfe (5/6): Intellireward Evaluation**
>
> > **How is the Intellireward evaluated? As far as I can tell, it seems the authors fine-tuned the LLM on the ProbeVote-500 dataset, and then conducted evaluation on the (training) data? Then it seems natural that the model can achieve way better performance compared to the other baselines reported in table 1.**
>
> IntelliReward is our reward model which we trained on the Train split of ProbeVote-500(Human annotation dataset) and evaluated it on the **eval split of ProbeVote-500**. All the Candidate Reward models were also evaluated on the eval set of ProbeVote-500 to ensure fairness.
>
> The below table shows alignment/agreement of more than 70% on each binary label(True, False) across 3 metrics(Grounding, Evidence, Effort). We have added the corresponding figure in Appendix (Figure 7 Section A.3).
>
> | Category   | Match on True Labels (%) | Match on False Labels (%) |
> |------------|---------------------------|----------------------------|
> | Effort     | 70                        | 72                         |
> | Evidence   | 80                        | 72                         |
> | Grounding  | 70                        | 75                         |
>
>
> In Below table (Same as Table 1 in paper) you can see the evaluation of various candidate reward models, calculated on eval split of ProbeVote-500
>
> | Candidate reward model | Checkpoint | Effort (%) | Evidence (%) | Grounded (%) | Mean Accuracy(%) |
> | :--- | :--- | :--- | :--- | :--- | :--- |
> | **Closed-source LLMs (off-the-shelf)** | | | | | |
> | Gemini 2.5 Flash | Original/API | 57 | 25 | 29 | 37 |
> | GPT-4.1 | Original/API | 44 | 22 | 30 | 32 |
> | GPT-5 | Original/API | 56 | 54 | 49 | 53 |
> | **Closed-source LLMs (tuned with SFT via API)** | | | | | |
> | Gemini 2.5 Flash | SFT/API | 61 | 53 | 45 | 53 |
> | GPT-4.1 | SFT/API | 52 | 25 | 31 | 36 |
> | **Open-source baseline** | | | | | |
> | Qwen2.5-7B-Instruct-1M | Original | 30 | 26 | 28 | 28 |
> | **Our trained reward model** | | | | | |
> | **IntelliReward (ours)** | – | **70** | **76** | **70** | **72** |

---

> ### Author Response · Authors · 2025-11-21
> **Response to Reviewer Xtfe (6/6): Dataset used to train IntelliAsk and Qwen2.5-7B SFT**
>
> > **Also the comparison between the IntelliAsk model and the Qwen2.5-7B SFT model seem to be not fair as they are trained on completely different datasets. Why not directly conduct SFT on the “preferred questions” rated by IntelliAsk and compare the model performances?**
>
> We believe there is a misunderstanding: **both IntelliAsk (LLM) and the Qwen2.5-7B SFT baseline are trained on exactly the same dataset, Probe-15K.**
>
> ProbeVote-500 is used to train the reward model (IntelliReward) which provides reward signals to IntelliAsk (LLM) during RL training. Probe-15k dataset contains questions that are written by human reviewers and were passed through multi-stage filtering with strict rules, therefore the dataset already contains very high quality questions.
>
> The table below shows the dataset used to train the Qwen2.5-7B SFT and our model IntelliAsk(7B and 3B)
>
> | Model                 | Training | Dataset    | Effort | Evidence | Grounding |
> |-----------------------|---------|------------|--------|----------|-----------|
> | Qwen2.5-7B            | Base    | -          | 0.00   | 0.01     | 0.01      |
> | Qwen 2.5-7B SFT       | SFT     | Probe-15k  | 0.00   | 0.01     | 0.02      |
> | IntelliAsk-7B | RL | Probe-15k  | 0.03   | 0.07     | 0.17      |
> | IntelliAsk-32B | RL | Probe-15k  | 0.23   | 0.12     | 0.20      |

---

### Author Response · Authors · 2025-12-03
**Summary of Comments and Experiments**

We thank the reviewers and the AC for their constructive feedback. We have conducted extensive additional experiments, scaled to **IntelliAsk-32B**, and revised the paper.

Below is a summary addressing **Human Annotation**, **Reward Model Design**, **Baseline Fairness**, and **Generalization**.

### 1. New Experiments: Generalization & Human Evaluation (IntelliAsk-32B)
To address concerns regarding generalization and potential "reward hacking", we trained **IntelliAsk-32B**. We performed **rigorous human evaluation** and tested the model on external benchmarks (DROP, MuSR, BoolQ, GPQA, etc.) and out-of-domain papers (ICML, NeurIPS, CVPR).

**A. Human Preference Evaluation**
Humans consistently preferred questions generated by IntelliAsk-32B over leading models.

| Competitor Model | IntelliAsk-32B Preference | Competitor Preference |
| :--- | :---: | :---: |
| **Gemini 2.5 Flash** | 81% | 19% |
| **GPT-4.1** | 85% | 15% |
| **Qwen3-32B** | 96% | 4% |

**B. External Benchmarks (Generalization)**
IntelliAsk-32B consistently outperforms the base model (Qwen3-32B), demonstrating that our training improves reasoning and writing capabilities beyond just the training task.

| Benchmark | IntelliAsk-32B | Qwen3-32B | Metric |
| :--- | :---: | :---: | :--- |
| **Reasoning & Comprehension** | | | |
| Eluther/DROP | **95.1** | 93.3 | F1/Acc |
| MuSR | **68.3** | 64.7 | Accuracy |
| GPQA-Diamond | **69.1** | 68.4 | Accuracy |
| **Writing & Generation** | | | |
| WritingBench | **8.31** | 8.07 | 0–10 |
| Arena Hard | **94.1** | 93.8 | 0-100 |
| **Domain Generalization** | | | |
| Conf. Mix '25 (ICML/NeurIPS) | **0.65** | 0.07 | Expected Score (0–3) |

---

### 2. Binary Annotation & Agreement Scores
We received questions regarding the choice of binary over Likert scales and the reliability of annotators.

**Why Binary?**
Pilot experimens (25% of annotation), using a 1-5 Likert scale revealed a **strong bimodal distribution**, where >85% of ratings were either 1 or 5. Annotators found intermediate scores ambiguous for short questions.

| Category | Score 1 | Score 2 | Score 3 | Score 4 | Score 5 |
| :--- | :--- | :--- | :--- | :--- | :--- |
| **Effort** | 64% | 5% | 1% | 2% | 28% |
| **Evidence** | 59% | 5% | 0% | 4% | 32% |
| **Grounded** | 61% | 3% | 1% | 5% | 30% |

**Inter-Annotator Agreement (IAA)**
We achieved strong agreement (Cohen's $\kappa$) across the final three attributes. We removed a fourth attribute, *Specificity*, during the pilot due to high variability ($\kappa = 0.59$).

| Source | Effort | Evidence | Grounding |
| :--- | :--- | :--- | :--- |
| **Human** | 0.92 | 0.88 | 0.90 |
| **Qwen2.5-32B** | 0.93 | 0.89 | 0.89 |
| **o3** | 0.84 | 0.86 | 0.87 |
| **Gemini 2.5 Pro** | 0.86 | 0.85 | 0.86 |

---

### 3. Reward Model Design and Evaluation
We addressed questions regarding our specific architectural choice (TransformerResidualHead) versus standard MLPs.

**(MLP vs. TransformerResidual)**
To empirically validate our design, we extended our experiments to include a **Trainable Base Model** comparison. As shown in the table below, while unfreezing the base model improves the standard MLP baseline ($62\% \rightarrow 67\%$), it still fails to match our proposed architecture.

Notably, our **Frozen Base + TransformerResidual** ($72\%$) outperforms the **Trainable Base + MLP** ($67\%$), confirming that the performance gain comes from our architectural design.

| Base Model | Pooling | Head Type | Mean Accuracy (%) |
| :--- | :--- | :--- | :--- |
| **Standard Reward Model** | | | |
| Frozen | None | MLP | 62 |
| Frozen | Pool50 | MLP | 65 |
| Trainable | None | MLP | 63 |
| Trainable | Pool50 | MLP | 67 |
| **IntelliReward (Ours)** | | | |
| Frozen | None | Transf. Resid. | 69 |
| Frozen | Pool50 | Transf. Resid. | **72** |
| **Trainable** | **Pool50** | **Transf. Resid.** | **73** |

*Note: Pool50 denotes mean pooling over the last 50 output tokens.*

**Evaluation on True/False Labels**
IntelliReward aligns with human judgment >70% of the time on the hold-out ProbeVote-500 evaluation set.

| Category | Match on True Labels (%) | Match on False Labels (%) |
| :--- | :--- | :--- |
| **Effort** | 70 | 72 |
| **Evidence** | 80 | 72 |
| **Grounding** | 70 | 75 |

---

### 4. Baseline Fairness (SFT vs. RL)
We clarified that **both** the SFT baseline and IntelliAsk were trained on the **exact same dataset (Probe-15k)**.
* **SFT Result:** The reward curve remained flat (see revised Fig. 5), indicating SFT alone fails to capture the nuance of high-quality reviewing.
* **RL Result:** IntelliAsk, utilizing signals from IntelliReward, showed a clear upward trend in reward and quality.

### 5. Paper Revisions
* Updated **Tables 1, 2, 3 & 4** with the new 32B results and extended ablations.
* Added bimodal Likert distribution plot (**Appendix A.2**).
* Added Inter-Annotator Agreement plot (**Appendix A.6**).
* Included UI setups and extended rejection sampling results (**Appendix A.4, A.10**).

---

### Note · Authors · 2026-01-06

I have read and agree with the venue's withdrawal policy on behalf of myself and my co-authors.